# OPENREVIEWER: MITIGATING CHALLENGES IN LLM REVIEWING

## ABSTRACT

Human reviews of research papers are slow and of variable quality. Hence there is increasing interest in using large language models (LLMs) such as GPT to review research papers. This paper develops a proof-of-concept LLM review process that shows LLMs offer consistently high-quality reviews almost instantly. However, many challenges and limitations remain: risk of misuse, inflated review scores, overconfident ratings, skewed score distributions, and limited prompt length. We mitigate these issues without prompt engineering by using LLM watermarking to mark LLM-generated reviews; classifying and detection errors and shortcomings of papers; and using long-context windows that include the review form, entire paper, reviewer guidelines, code of ethics and conduct, area chair guidelies, and previous year statistics; and a blind human evaluation of reviews. We aim to use OpenReviewer to review and revise research papers, improving their quality. This work identifies and addresses drawbacks associated with GPT as a reviewer and enhances the quality of the reviewing process based on a randomized human blind evaluation. Making OpenReviewer available as an open online service that generates reviews will allow the use of scalable human feedback to learn and improve.

## 1 INTRODUCTION

Reviewing research papers is a fundamental aspect of scientific research; however, it entails lengthy waiting months for human experts to complete reviews of variable quality. Large language models (LLMs) such as OpenAI's GPT-4 (OpenAI, 2023) have the potential to augment traditional reviews by providing instant reviews that are high-quality in several aspects and have lower variance compared to human reviews. However, applying LLMs in this context has challenges, including risk of misuse, inflated review scores, overconfident ratings, skewed score distributions, and limited prompt length. This paper presents approaches to address these challenges and a proof-of-concept LLM review tool that demonstrates the potential of LLMs to provide immediate, high-quality feedback on many aspects of research papers.

We provide the acceptance and rejection rates, scores, and content analysis of ICLR papers submitted to OpenReview.net, both with and without concurrent arXiv.org submissions. This analysis also informs the LLM reviewer about the acceptance rates of papers, enabling it to generate more informed and contextually appropriate reviews. We introduce an LLM watermarking procedure for closed-source LLMs to identify reviews generated by an LLM. While previous watermarking efforts have focused on open-source LLMs, our approach allows us to work with closed models. We explore using extended context windows in the review process, incorporating, in addition to the review forms, the reviewer guidelines, codes of ethics, and conduct. An ablation study identifies the impact of including these elements on LLM review quality, providing insights into how to improve the reviews.

We categorize various types of errors found in ICLR papers and subsequently introduce these errors into papers. OpenReviewer then reviews these papers, both in their original and error-introduced forms. We compare the reviews of the unaltered papers with those of the error-introduced papers. Finally, we identify the errors in the review text of the papers with the errors and their corresponding scores. We further sub-categorize theoretical and technical errors into more specific types and differentiate between errors that can and cannot be detected.

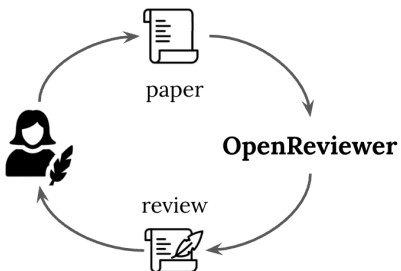

Figure 1: An author or reviewer (or area chair) who receives an LLM review may improve their paper or review quality. The author or reviewer submits the paper to OpenReviewer, automatically receives a review, learns, and improves the paper or their review (or meta-review) before actual submission or review (or meta-review).

We conduct a blind human evaluation, comparing reviews generated by three humans and three LLM variants. This paper also identifies valid and invalid uses of LLM reviews. While LLMs effectively improve review presentation quality, they could be more adept at judging the correctness of mathematical content. We propose a test procedure to identify improvements as LLMs evolve.

Finally, we introduce an open online service, OpenReviewer [1], to which users upload papers and receive back reviews within minutes. This service enables continuous improvements and provides a platform for scalable human feedback on the quality and usefulness of LLM reviews. Figures 1 and 2 illustrate a high-level learn and improve loop and the OpenReviewer architecture, highlighting contributions in green.

**Related work.** LLMs demonstrate surprising capabilities evaluating texts, though generate hard-to-detect errors such as hallucinations and, show a remarkable power to persuade humans even when inaccurate (Spitale et al., 2023). This makes controlling the quality and appropriateness of LLM-augmented reviewing highly challenging. The academic community acknowlegdes the acute need for LLM-augmented reviewing (Liu & Shah, 2023; Robertson, 2023; Petrescu & Krishen, 2022; Schulz et al., 2022; Checco et al., 2021; Bao et al., 2021; Vesper, 2018), along with the risks (Kaddour et al., 2023; Spitale et al., 2023; Zou et al., 2023). Previous work questions LLMs ability to perform reviewing sub-tasks Liu & Shah (2023).

## 2 METHODS

### 2.1 DATASET

We curated a dataset of 4,956 papers with 18,565 reviews from ICLR 2023 publicly available on OpenReview.net OpenReview (2023). Of the papers publicly available on OpenReview.net, 2,040 (53.3%) are also available on arXiv.org arXiv (2023) with a total of 7,698 reviews (41.5%) as shown in Figure 6. We extracted the directories of each arXiv paper, detected their main text file, removed text before the paper title, and replaced any imported input text with the text itself, curating 2,040 full papers. In addition to the papers and their reviews, we collected the statistics of the decisions and scores of ICLR 2022, the ICLR 2023 reviewer guide, code of ethics, code of conduct, and review form. Appendix A provides paper and review statistics.

### 2.2 LLM REVIEWING

In our efforts to understand the capabilities of LLMs in the peer review process, we conducted a series of ablation studies, each introducing more context and information to the LLM. These ablation studies were done on a random sample of papers representing 10% of our dataset. This subset of data maintained the same distribution of accepted types and rejected papers as the broader dataset.

---

[1]openreviewer.com

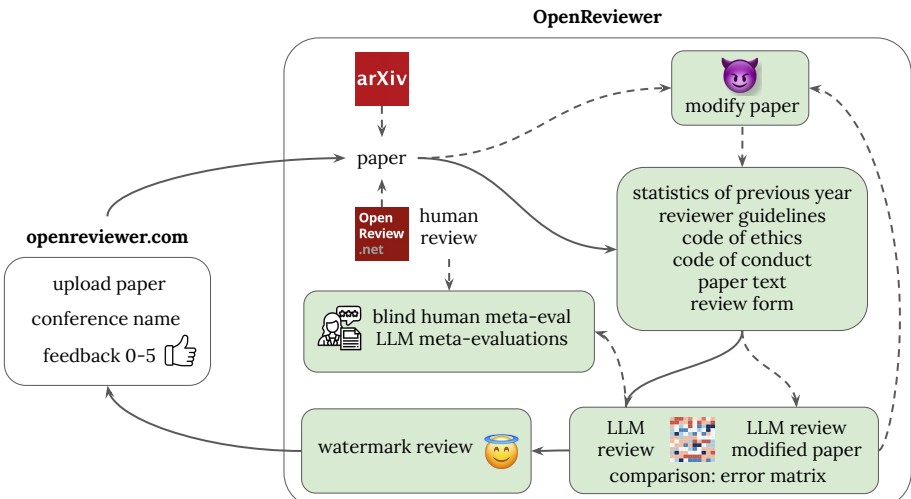

Figure 2: OpenReviewer architecture: During inference (solid lines), the authors upload a paper with the conference name. The paper is reviewed by OpenReviewer using GPT-4 with the conference statistics of the previous year, reviewer guidelines, code of ethics, code of conduct, full paper text, and review form. The LLM review is then watermarked and returned to the author, who provides feedback about the review. The paper is deleted from the system. Offline (dashed lines) conference papers and reviews are downloaded from OpenReview.net and arXiv.org. Conference statistics of the previous year, reviewer guidelines, code of ethics, code of conduct, full paper text, and review form are curated. GPT-4 is used to review the papers. A blind meta-evaluation is performed on a sample of human and LLM reviews by human experts who are senior/area chairs of the conference and used to evaluate the LLM review quality. The papers are modified by introducing errors or shortcomings, and the LLM reviews the original and modified papers. The original paper reviews are compared with the modified paper reviews for score, and the content is analyzed by detecting the modifications. This identifies which types of errors and shortcomings the LLM is sensitive to in its review and which types it cannot reliably review, defining the review limitations.

We ablate context by increasing the documents used as shown in Table 1. Context P1 includes the full paper text (P) and conference review form (RF). P2 adds the reviewer guide (RG). P3 adds the code of ethics (CE) and code of conduct (CC). P4 adds guidelines for the area chairs (AC). P5 adds the statistics of the previous years conference. Appendix B provides example paper LLM reviews.

To ensure the reliability of LLM's review quality it is crucial to conduct a thorough evaluation by human experts. We perform review evaluator-review writer evaluations for human-human, human-LLM, LLM-human, and LLM-LLM evaluations, as well as meta-evaluations of humans of the LLM-LLM evaluations.

## 2.3 EVALUATING REVIEWS

Evaluating LLMs is challenging. We tackle this challenge by having the LLM evaluate reviews of humans and LLMs and using blind human evaluation of the human and LLM reviews. The human evaluation of the human and LLM reviews is performed by three experts in the field, including researchers who have served as area and senior area chairs. Human expert evaluation of LLM and human reviews is performed by:

- Randomly shuffling the papers and choosing 10% of the papers randomly.
- Randomly permuting each of the reviews for each paper, which consists of three human reviews and three LLM reviews.
- Blindly evaluating the reviews by human experts. The experts do not know whether the review being evaluated was written by a human or an LLM, and each expert answers four meta-questions about each review.

| Document | Description |
|---|---|
| Statistics (S) | Histogram and statistics of reviewer scores and recommendations for papers in previous year (ICLR 2022). |
| Area Chair (AC) Guidelines | Aiming for a percentage range of papers acceptance rate and range of paper scores that generally meet this rate. Papers can be accepted or rejected outside this range, but reasons must be given. |
| Reviewer Guide (RG) | Responsibilities and expectations for reviewers, step-by-step reviewing instructions, review examples, and FAQs. |
| Code of Ethics (CE) | Promotes responsible research in machine learning and AI, emphasizing societal contribution, scientific excellence, harm avoidance, honesty, fairness, respect for work, privacy, and confidentiality. |
| Code of Conduct (CC) | Promotes open exchange of ideas, respect, and freedom of expression. It prohibits harassment, bullying, discrimination, and retaliation. |
| Review Form (RF) | Sections for summarizing the paper, evaluating its strengths and weaknesses, correctness, rating its technical and empirical novelty, assessing any potential ethical issues, overall recommendation, and confidence in that recommendation. |
| Paper (P) | Entire text including inputs |

Table 1: Documents for review and their descriptions.

The four meta-questions answered by the three human experts for each review, along with instructions that ensure calibrated evaluations are:

1. "How well does the review explain the score?"
   Instructions: On a scale of 0-5 (0 being the worst, 5 the best), evaluate how well does the review summary, strengths and weaknesses, questions, and limitations explain the overall number rating.

2. "How well does the review guide the authors to improve the paper?"
   Instructions: On a scale of 0-5 (0 being the worst, 5 the best), evaluate how useful are the review strengths and weaknesses, especially questions and limitations for improving the paper.

3. "Does the review contain content specific to the paper?"
   Instructions: On a scale of 0-5 (0 being the worst meaning content free, and 5 being the best meaning that the review is based on content), evaluate if review summary, strengths and weaknesses, questions, and limitations are based on content.

4. "For each paper, which three of reviews R1,R2,R3,R4,R5,R6 do you think are generated by an LLM?"

To validate that the expert human evaluation is blind, we ask the human expert evaluator to guess which three of the reviews, permuted in random order, are by an LLM. The human expert guessed correctly 59.8% of the time (random is 50%), which validates that the evaluation process is blind. By conducting these evaluations, we gain a deeper understanding of the LLM's performance and ensure that its reviews meet the required standards of quality and accuracy.

## 2.4 WATERMARKING

We use an LLM watermark Kirchenbauer et al. (2023) that is easy to detect. We watermark the review summary by rephrasing it using a set of words from the vocabulary, which modifies the distribution of words while maintaining the summary quality. We are aware that this watermark may be tampered with by applying the reverse process of rephrasing the watermarked summary again using a closed LLM.

## 2.5 ERROR AND SHORTCOMING DETECTION

We modify the papers by introducing errors or shortcomings, and an LLM reviews the original and modified papers. The original paper reviews are compared with the modified paper reviews for score, and the content is analyzed by detecting the modifications. This identifies which types of

errors and shortcomings the LLM is sensitive to in its review and which types it cannot reliably review, defining the review limitations.

| Error or Shortcoming | Description |
|---|---|
| Theoretical Mistakes | These can range from incorrect mathematical derivations to making unfounded assumptions about a model or algorithm. |
| Metrics | Not reporting important metrics or details about experiments. |
| Related Work | Not comparing with the state-of-the-art or relevant baselines. |
| Overclaiming | Making exaggerated claims about the novelty or impact of the work without substantial evidence can be problematic. |
| Insufficient Ablation Studies | Ablation studies help demonstrate which components of a proposed system contribute to its performance. Without these, it can be hard to understand the significance of the introduced changes. |
| Lack of Baseline Comparisons | Not comparing with standard or widely-accepted baseline methods can contribute to seeming less grounded. |
| Ethical Concerns | Not addressing potential ethical implications of the work, especially if the work touches on sensitive areas like facial recognition, can be a red flag. |
| Lack of Discussion on Limitations | Every approach has limitations. Not discussing them or addressing potential criticisms can show a lack of thoroughness. |
| Citation Issues | This includes not citing relevant prior work, which can make it seem like the authors are unaware of the literature or over-citing one's previous work without reason. |
| Technical Errors | These can range from coding mistakes in the provided implementation to incorrect use of statistical tests. |

Table 2: We classify different types of errors in papers and then introduce these errors into papers. OpenReviewer reviews the papers without and with the errors. We compare the reviews of the original papers and the reviews of papers with the errors, and finally, we detect the errors in the text of the review of the papers with the errors and their scores. Theoretical mistakes and technical errors are further classified into sub-types.

## 2.6 MITIGATING MISUSE

During this work we devised preventive actions that are particularly urgent for LLM reviewing summarized in Table 3.

| Action | Description |
|---|---|
| Declaration | Authors and reviewers should declare when using an LLM to ensure transparency. |
| Self-regulation | The LLM should self-prompt to check for harmful, biased, or unaligned content. This can be done through a two-step approach where the LLM evaluates its output before responding to the user. |
| Gatekeeping checklist | The same guidelines and regulations for human reviewers should be applied to machine reviews. This includes a mandatory checklist of questions for the human and machine reviewers flagging ethics, adhering to reviewer duties, and reviewer confidence. |
| Adherence to the conference code of conduct | Both human and machine reviewers should abide by the same code of conduct. This includes following the exact gate-keeping mechanisms, alerts when breaking the rules, and regulations by editors and professional associations. |
| Debiasing | Identify bias by examining evaluations against unbiased benchmarks, identify non-representative reviewer characteristics, and regularize by fairness criteria. |
| Explanations | Deeper explanations are needed to validate LLM reviews. These can be solicited, for example, using chain-of-thought prompting. Quality control should be done before running the machine and ensure correlation with benchmarks. This involves self-reflection of the LLM to help control delegation and mitigate misalignment of objectives and information asymmetry. |

Table 3: Preventive actions for ethical and transparent use of LLMs in the peer review process.

## 3 RESULTS

Initially this paper received a recommendation score of 5 by OpenReviewer. Subsequently, we addressed the weaknesses, and improved, receiving a recommendation score of 7.

### 3.1 REVIEW SCORES

Table 4 shows the average and standard deviation scores of the human reviewers and LLM review for paper correctness, technical novelty and significance, empirical novelty and significance, overall recommendation score, and confidence. P1, P2, P3, P4, P5 ablate the increasing documents used in the GPT-4 context prompt. P1 includes the full paper text (P) and conference review form (RF). P2 adds the reviewer guide (RG). P3 adds the code of ethics (CE) and code of conduct (CC). P4 adds guidelines for the area chair (AC). P5 adds the statistics of the previous year's conference.

The human reviewers have an average recommendation score of 5.88, with a standard deviation 1.61. With the context of the entire paper text and the conference review form (P1), the LLM has an average recommendation of 7.21, higher than the human reviewers. The standard deviation of 1.03 is less than that of the human reviewers. Adding the reviewer guide to the context (P2) slightly increases the recommendation score to 7.58. The standard deviation is reduced further, with a more consistent scoring by the LLM. With the addition of the code of ethics and code of conduct to the context (P3), the recommendation score slightly increases to 7.62, similar to P2, and the standard deviation remains the same. After adding guidelines for the area chair (P4), the recommendation score decreases to 4.61, indicating that this context makes the LLM more critical or stringent in its evaluations due to the knowledge of expected outcomes. With the addition of the previous year's conference statistics (P5), the recommendation score improves and is near the human reviewer's score. The standard deviation also increases, indicating more variability in the scoring. In summary, LLM contexts P1, P2, and P3 consistently give higher recommendation scores than the human reviewers, suggesting a more positive or lenient view of the papers. P4 context, with the area chair guidelines added, shows a significant decrease in recommendation scores, suggesting these guidelines influence the LLM to be more critical in its evaluations. P5 reaches the same level of recommendation as the human reviewers.

| Reviewer | Correctness | Tech. Novelty | Emp. Novelty | Rec. | Confidence |
|---|---|---|---|---|---|
| Human | 3.18±0.64 | 2.62±0.72 | 2.66±0.73 | 5.88±1.61 | 3.67±0.71 |
| P1: P+RF | 3.07±0.37 | 3.16±0.47 | 3.16±0.43 | 7.21±1.03 | 3.80±0.47 |
| P2: RG+P+RF | 3.06±0.24 | 3.16±0.36 | 3.17±0.40 | 7.58±0.65 | 3.84±0.41 |
| P3: RG+CE+CC+P+RF | 3.03±0.20 | 3.15±0.37 | 3.13±0.36 | 7.62±0.65 | 3.86±0.44 |
| P4: AC+RG+CE+CC+P+RF | 2.74±0.48 | 2.56±0.53 | 2.56±0.52 | 4.61±0.83 | 3.86±0.38 |
| P5: S+AC+RG+CE+CC+P+RF | 2.81±0.57 | 2.84±0.66 | 2.84±0.60 | 5.56±1.43 | 3.84±0.44 |

Table 4: Ablation of in-context LLM review scores: Average and standard deviation scores of the human reviewers and LLM review for paper correctness, technical novelty and significance, empirical novelty and significance, overall recommendation score, and confidence. P1, P2, P3, P4, P5 ablate the increasing documents used in the GPT-4 context prompt. P1 includes the full paper text (P) and conference review form (RF). P2 adds the reviewer guide (RG). P3 adds the code of ethics (CE) and code of conduct (CC). P4 adds guidelines for the area chairs (AC). P5 adds the statistics of the previous years conference.

### 3.2 EVALUATING REVIEWS

The human review evaluator assesses reviews written by human reviewers and the LLM, GPT-4 with context P5. The human review writer is an ICLR 2023 reviewer. Table 5 shows the average evaluation results on a randomized sample of 5% of the papers evaluated by human experts.

Considering the average human review rating as ground truth, we perform an analysis of false positives and negatives, considering the LLM's two types of errors: accepting a paper rejected by human reviewers and rejecting a paper that human reviewers accepted. One paper that the LLM accepted with a score of at least 7 was rejected by the average of the human reviewers with a score of at most 3. Four papers that the LLM rejected with a score of at most 3 were accepted by the human

| Review Evaluator: | Human | Human | GPT-4 | GPT-4 |
|---|---|---|---|---|
| **Review Writer:** | **Human** | **P5** | **Human** | **P5** |
| How well does the review explain the score? | 4.80±0.39 | 4.76±0.51 | 4.27±0.65 | 4.65±0.52 |
| How well does the review guide the authors to improve the paper? | 4.66±0.51 | 4.79±0.71 | 4.14±0.50 | 4.27±0.45 |
| Does the review contain content specific to the paper? | 4.53±0.79 | 4.68±0.82 | 4.97±0.16 | 4.95±0.22 |

Table 5: The human review evaluator evaluates human and P5 written reviews of papers. The human review writer is an ICLR 2023 reviewer. The LLM is GPT-4 with context P5. The evaluation is on a scale of 0-5 (0 being the worst, five the best). For the third question, a score of 0 indicates a content-free review.

reviewers with a score of at least 7. Eight papers that the LLM accepted with a score of at least 6 were rejected by the average of the human reviewers with a score of at most 4. 22 papers that the LLM rejected with a score of at most 4 were accepted by the human reviewers with a score of at least 6.

### 3.3 Watermarking

Figure 3 shows a comparative analysis of token distribution and "green list" token presence between original and watermarked summaries. The top-left histogram illustrates that watermarked summaries, represented in red, consistently contain more tokens than the original ones. The top-right histogram reinforces this by indicating that a greater fraction of "green list" tokens is present in the watermarked summaries. The box plots in the bottom row of Figure 3, show that watermarked summaries consistently have a higher median token count and a greater fraction of green list tokens. Additionally, a wider interquartile range for both metrics in watermarked summaries suggests greater variability.

### 3.4 Errors and Shortcomings

Figure 4 shows the average scores for each error or shortcoming type: The human review average score' serves as a reference, while the error types are ranked. Figure 5 shows a heatmap of scores of each error or shortcoming type. The color gradient, ranging from blue (low score) to red (high score), indicates the score magnitude.

### 3.5 Limitations

We distinguish between the limitations of our research methodology and OpenReviewer, our application of GPT-4 to reviewing papers in computer science. The closed nature of GPT-4 restricts our choice of methodology, which includes an iterative design process coupled with a set of evaluation methods. Unfortunately, there is no tested evaluation framework that we could apply a priori, and we must resort to gradually refining the evaluation method. Designing OpenReviewer is an iterative process that tests alternative information sources, such as conference guidelines and knowledge of past editorial decisions, various reviewing criteria, such as novelty, correctness, and ethical considerations, and the corresponding errors and shortcomings of the submission to be detected by the reviewer. This highly work-intensive approach is a nuanced method that relies on trial-and-error attempts to calibrate the tool. It would have to be repeated for conferences in other domains or other ways of evaluating academic publications. Furthermore, as we learn to use LLM reviewing and as LLMs develop further, we will have to reevaluate the application design continually. For this reason, we make OpenReviewer available as a beta demonstration to collect user feedback on a larger scale, to learn and improve.

The evaluation of GPT reviewing relied on evaluating the reviews rather than the process. We used both LLM evaluations and human evaluations of the reviews, and both concluded that the quality of the GPT-4 review matched that of humans but could not compare the processes other than to note the reviewers' confidence in their conclusions. Future research should extend our evaluation to examine how authors, for example, use and trust OpenReviewer reviews.

The limitations of OpenReviewer are one of the contributions of our paper. Our analysis of its capabilities by classifying and testing various reviewing criteria and types of errors and shortcomings

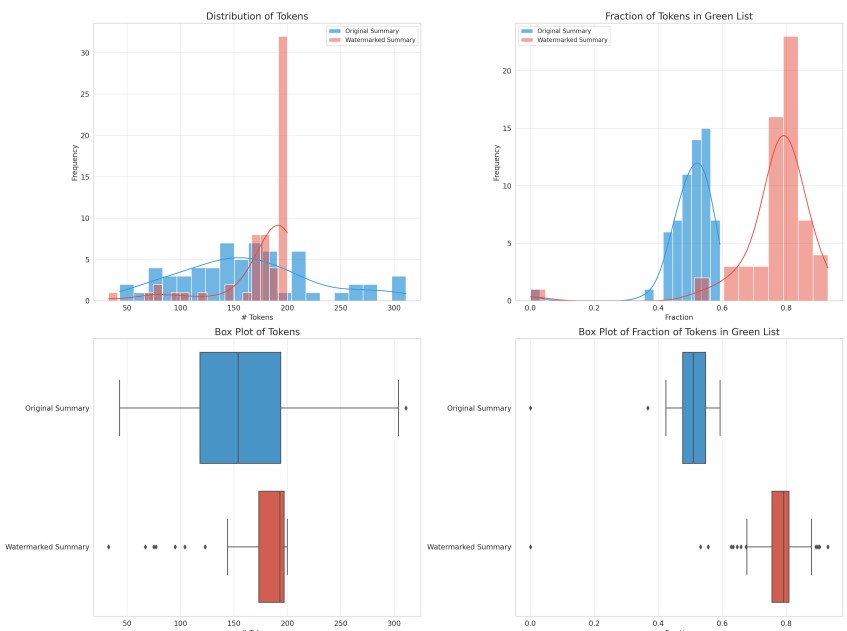

Figure 3: Comparative analysis of original vs. watermarked summaries: Side-by-side comparison of token distributions and the fraction of tokens appearing in a predefined "green list" for original and watermarked summaries. On the top left, the histogram illustrates the number of tokens in summaries, with the original summaries shown in blue and the watermarked in red. The top right histogram contrasts the fraction of tokens from the green list present in each summary type. The bottom row's box plots shows these metrics' spread and central tendencies. Watermarked summaries consistently have a higher token count and a greater fraction of tokens from the green list. We use OpenAI's GPT-4 to generate the original review summaries, and Meta's OPT-7.6B to rephrase the summaries and apply the LLM watermark.

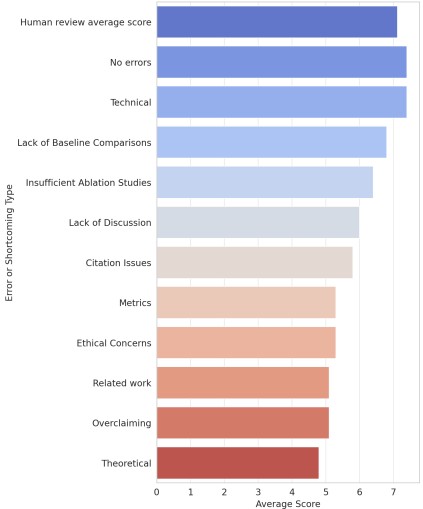

Figure 4: Average scores for each error or shortcoming type: The human review average score' serves as a reference, while the error types are ranked.

indicates the limits of our current application. These limitations are essential for knowing how to use the application, particularly involving the human reviewer.

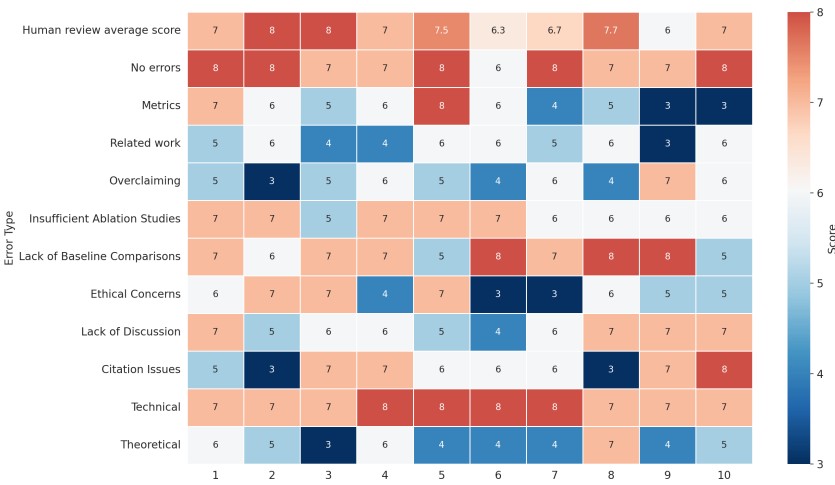

Figure 5: Heatmap of scores of each error or shortcoming type. The color gradient, ranging from blue (low score) to red (high score), indicates the score magnitude.

## 4    CONCLUSIONS

Following Bommasani et al. (2021), we examine the capabilities, opportunities, and risks of using LLMs in reviewing academic papers. Our working assumption is that at this early stage of employing LLM for reviewing, it is necessary to learn from doing about the opportunities and, all the more so, risks of LLM reviewing. The rapid developments of LLM capabilities require continual monitoring and reexamining of opportunities and risks that build on previous rounds. To our knowledge, we are the first to report on a large empirical evaluation of LLM reviewing. The underlying evaluation method may contribute to subsequent evaluations that will follow with the development of new LLM capabilities. Moreover, at this stage, our goal is to design the application of LLM to reviewing to match the quality of human reviewing. If and when LLMs performs better than human reviewing in certain aspects or scientific domains, the evaluation methods will change.

Using human reviews as a base line, our goal was to evaluate the value alignment and the process alignment of LLM reviews, i.e., to compare the quality of reviews and the adherence of the reviewing process to conference guidelines and scientific norms of practice. In particular, conference reviewing has two goals: evaluating submissions to determine an accept or reject decision and providing constructive feedback to the authors. Our evaluation of review quality addresses both aspects. Furthermore, prior work on LLM academic capabilities suggests that LLMs are now ripe for specific reviewing tasks and appear to be more effective for some academic domains and less effective for others Liu & Shah (2023); Schulz et al. (2022); Checco et al. (2021). Therefore, we conducted ablation studies to determine which information sources effectively detected errors and shortcomings.

Only when supplied with information about previous editorial decisions did the LLM achieve good alignment with human reviewers. Furthermore, the LLM performs well in detecting certain errors and shortcomings but not others, such as detecting cases in which the authors needed to follow expected norms. Our conclusions regarding the evaluation method have longer-term implications. For limited model transparency, iterative design and large-scale empirical evaluation proved essential to calibrate the application of LLMs.

In summary, this work leverages LLMs in the review process, addressing challenges and offering a proof-of-concept LLM review tool. We hope that our work paves the way for more efficient, consistent, high-quality reviews, accelerating scientific progress while maintaining responsible conduct of research.

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

ETHICS STATEMENT

In developing and applying our LLM review process, we followed the ICLR Code of Ethics and Conduct, prioritizing responsible research, transparency, and fairness. Using LLMs in the review process could potentially lead to harmful insights or applications if misused. An over-reliance on LLMs could marginalize human reviewers, leading to a loss of diverse perspectives in the review process. To mitigate this, we emphasize that our LLM reviewer is intended to augment, not replace, human reviewers, and we implemented a watermarking procedure to distinguish LLM-generated reviews from human reviews. We are aware that the LLM watermark may be detected and removed by rephrasing the review using another LLM, however we believe that making the methodology open, rather than closed, improves usage safety in the long-term.

We are aware of potential discrimination, bias, or fairness concerns. LLMs, including our reviewer, are trained on large datasets that may contain biased information. More importantly, we do not recommend using our LLM reviewer to make important academic decisions, but rather view it as a tool for authors to use to revise papers.

Privacy and security issues are important in our work. Our LLM reviewer [2] does not store the papers it reviews, ensuring the confidentiality of authors' work, however it does share the paper with OpenAI under their terms of service. Our open online service is designed with security measures to protect users' information. We only retain uploaded papers and review data for the period necessary to fulfill the service, after which the data is deleted. However, we cannot guarantee absolute security and are not responsible for any unauthorized access or hacking events. Users are responsible for ensuring they hold the copyright or permission for any material they upload to our service.

Finally, we uphold research integrity by being transparent about our methodologies and findings. We have conducted an evaluation of our LLM reviewer, including a blind human evaluation, and have reported our results honestly and accurately. We acknowledge the limitations of our work and are committed to continuous improvement of research based on feedback from users and the machine learning community. In conclusion, when used responsibly and ethically, our LLM reviewer can contribute positively to the research community by providing instant, high-quality reviews in several aspects of the review. We are committed to ongoing ethical reflection and adjustment as we develop and refine this new reviewing tool.

## A  PAPER AND REVIEW STATISTICS

**Review Decisions.**  Percentages of papers accepted or rejected comparing papers on OpenReview.net and papers also available on arXiv.org are shown in Figure 7. The distribution of the different decisions for the papers is shown in Figure 8.

- Acceptance Rates: Papers on OpenReview.net had varying acceptance tiers. Notably, 91 papers (2.4%) were recognized as top 5% notable papers, while 280 (7.3%) were designated as top 25% notable. In contrast, for the subset of papers also on arXiv.org, 67 papers (3.3%) were considered top 5% notable, and 193 papers (9.5%) as top 25% notable. This indicates that papers on arXiv.org had a slightly higher representation in the notable categories.

- Poster Acceptance: Of the total submissions on OpenReview.net, 1,202 (31.4%) were accepted as posters. For papers also available on arXiv.org, 795 papers (39%) were accepted as posters, indicating that arXiv papers had a slightly higher acceptance rate for poster presentations.

- Rejection Rates: A notable 2,235 papers (58.4%) on OpenReview.net were rejected, while this rejection rate dropped to 33.6% of 685 rejected papers also on arXiv.org. This stark difference implies that papers available on arXiv.org had a significantly lower rejection rate.

- Desk Rejections/Withdrawals: On OpenReview.net, 1,148 (30%) of the submissions were either desk rejected or withdrawn, compared to a lower 14.7% of 300 papers also on arXiv.org.

---

[2]openreviewer.com

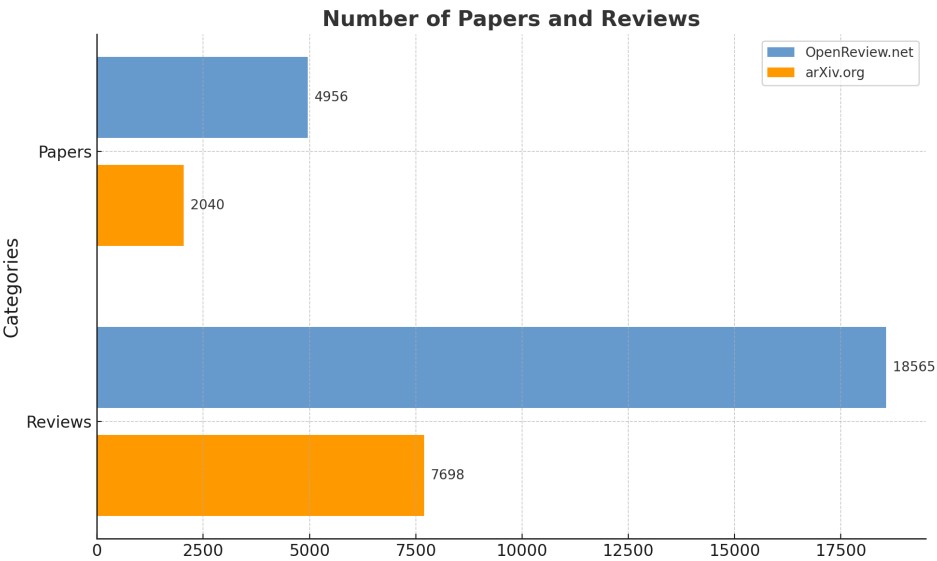

Figure 6: Number of ICLR 2023 papers and reviews publicly available on OpenReview.net and also available on arXiv.org.

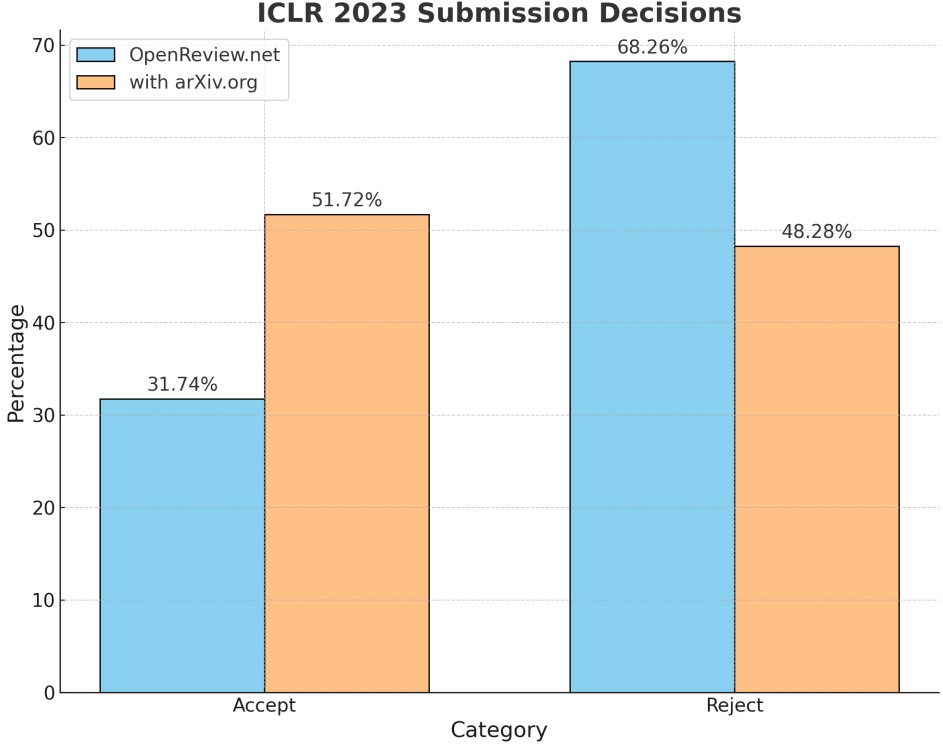

Figure 7: Papers submitted to ICLR, NeurIPS, and ICML through OpenReview.net are publicly available online. ICLR 2023 submission accepted less than a third of the papers submitted publicly to OpenReview.net, and more than half of the papers that were also publicly available on arXiv.org.

**Number of Reviews.** The distribution of the number of reviews each paper received is shown in Figure 9. The majority of papers on OpenReview.net received either 4 or 3 reviews, with 3,063

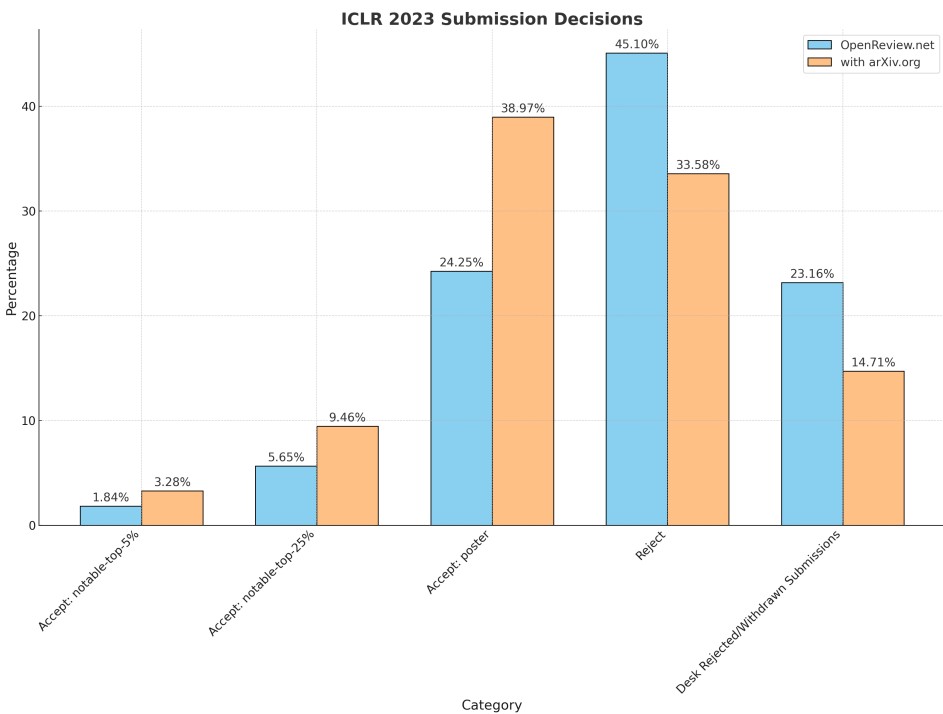

Figure 8: Distribution of paper decisions for papers on OpenReview.net and for papers also available on arXiv.org.

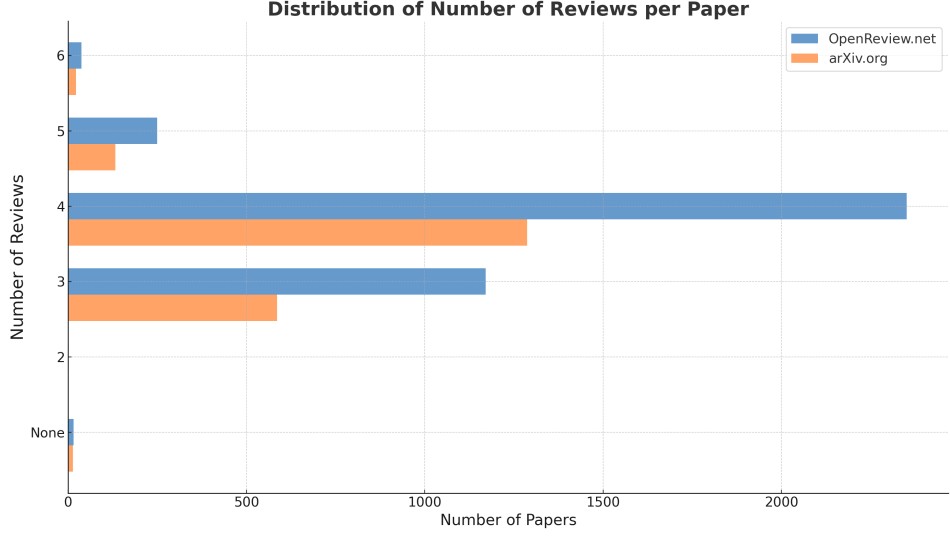

Figure 9: Distribution of the number of reviews for papers on OpenReview.net and for papers also available on arXiv.org. Most papers received either 4 or 3 reviews.

(30.35%) and 1,504 (61.8%) papers falling into these categories, respectively. The distribution was similar for papers also available on arXiv.org.

**Review Scores.** Figures 10, 11, and 12 present the distributions of review scores concerning the correctness, technical novelty and significance, and empirical novelty and significance, respectively. Generally, most papers received scores indicating some minor issues in claims or some overlap with

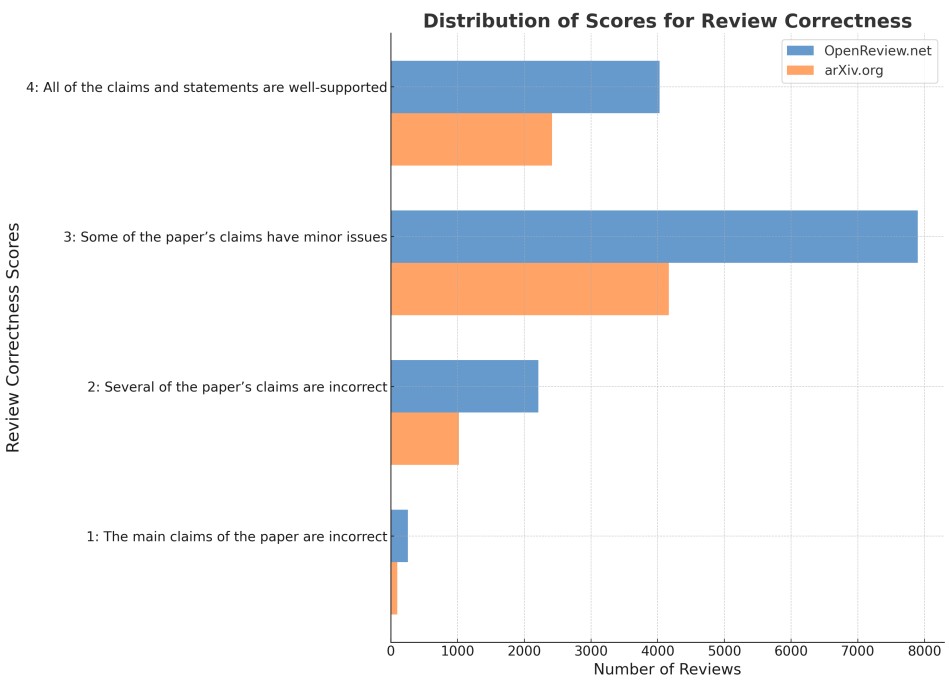

Figure 10: Distribution of human review scores for correctness for papers publicly available Open-Review.net and for papers also available on arXiv.org. A significant number of reviews indicate minor issues in the paper's claims.

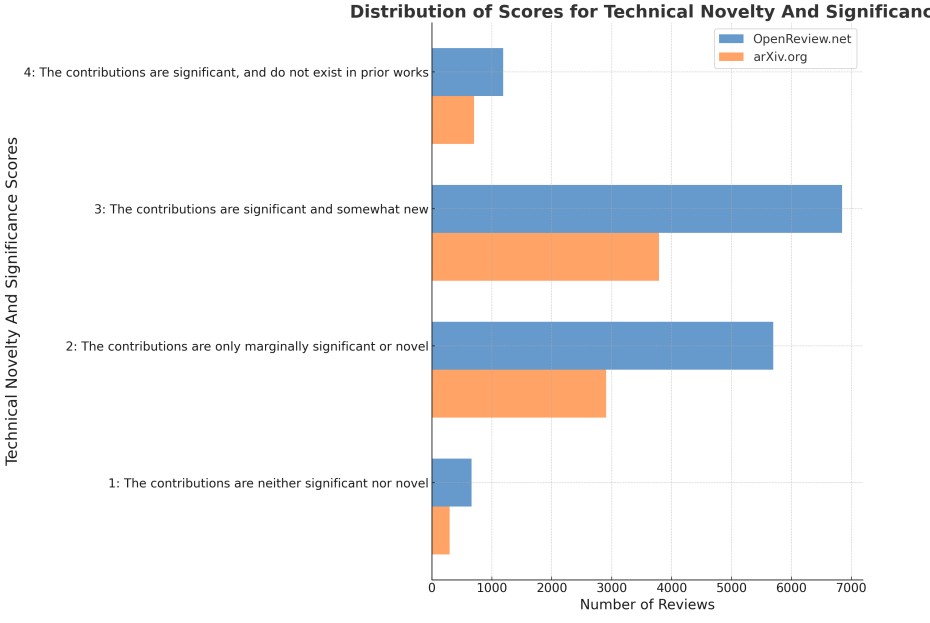

Figure 11: Distribution of human review scores for technical novelty and significance on Open-Review.net and for papers also available on arXiv.org. Many papers had contributions that were significant but had some overlap with prior work.

prior work. However, many of the papers had their claims and contributions rated as well-supported and significant.

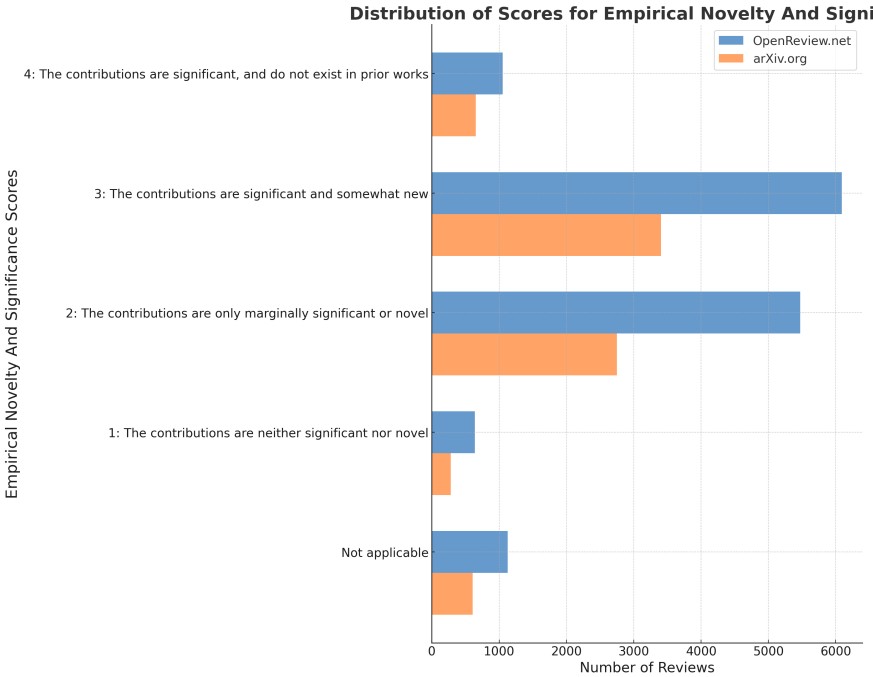

Figure 12: Distribution of human review scores for empirical novelty and significance for papers publicly available on OpenReview.net and for papers also available on arXiv.org. Many papers had contributions that were significant with some aspects existing in prior work.

- Review Correctness: On both platforms, most papers received scores indicating minor issues in their claims. However, a marginally higher proportion of papers on arXiv.org received scores indicating all claims were well-supported (59.1% on arXiv vs. 28% on OpenReview.net).

- Technical Novelty and Significance: Most reviews on both platforms suggested that the contributions of papers were significant but had some overlap with prior work. Notably, a slightly higher percentage of papers on arXiv.org (18.3%) were scored as having significant contributions that did not exist in prior works compared to OpenReview.net (8.3%).

- Empirical Novelty and Significance: Similar to technical novelty, many papers on both platforms had significant contributions, with some aspects existing in prior work. However, 20.1% of papers on arXiv.org were scored as having significant empirical contributions that did not exist in prior works, versus 7.3% on OpenReview.net.

**Review Recommendation and Confidence.** The distribution of review recommendations is shown in Figure 13. Most reviews fell into the categories of being marginally above or below the acceptance threshold. Figure 14 shows the distribution of review confidence scores. Notably, many reviewers were confident in their assessments but not certain.

- Review Recommendation: Most reviews on both platforms fell into the categories of being marginally above or below the acceptance threshold. However, a slightly higher proportion of papers on arXiv.org (34.6%) were recommended for acceptance compared to OpenReview.net (19%).

- Review Confidence: Many reviewers on both platforms were confident in their assessments, indicating a robust review process. For papers on arXiv.org, a slightly higher percentage of reviewers (12.3%) were absolutely certain about their assessments than OpenReview.net (10%).

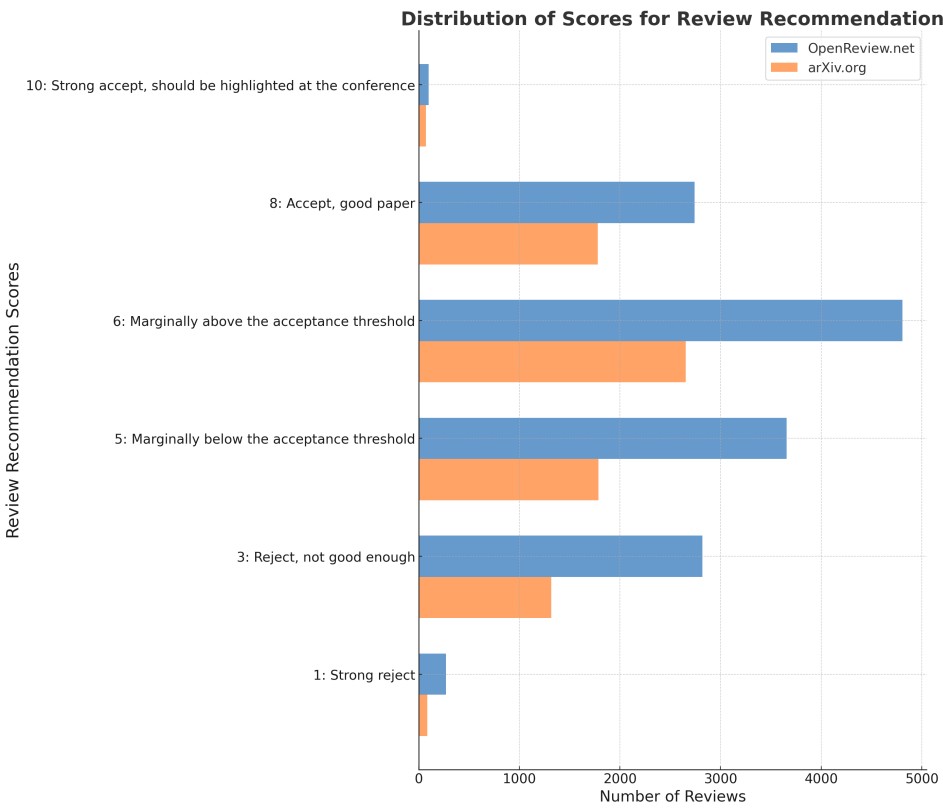

Figure 13: Distribution of human review recommendation scores for papers publicly available on OpenReview.net and arXiv.org: Ranges are from 1 to 10, with 10 being a strong accept and 1 being a strong reject.

Papers on OpenReview.net and arXiv.org are publicly available online. The data suggests that papers available on both OpenReview.net and arXiv.org generally receive more favorable reviews and decisions than those only on OpenReview.net.

Figure 16 shows the distribution of the lengths of the full papers available on arXiv. The mean is 11,298 words, and the median is 9,995 words. GPT-4 accepts a maximum of 32,000 tokens, about 24,000 words. The ICLR 2023 reviewer guide consists of around 3,200 words, and the review form is around 800 words, leaving 20,000 words for the paper and review. The mean human review length is less than 500 words, as shown in Figure 17. This leaves about 19,500 words for the entire paper. 147 of the papers (7.2%) are longer than this length. Including the ICLR code of ethics (1,770 words) and code of conduct (663 words) leaves about 17,000 words for the paper. 237 papers (11.6%) are longer than this length. GPT-4 has a context window of size 32k tokens that are suitable for handling around 90% of the full papers, including all related documents.

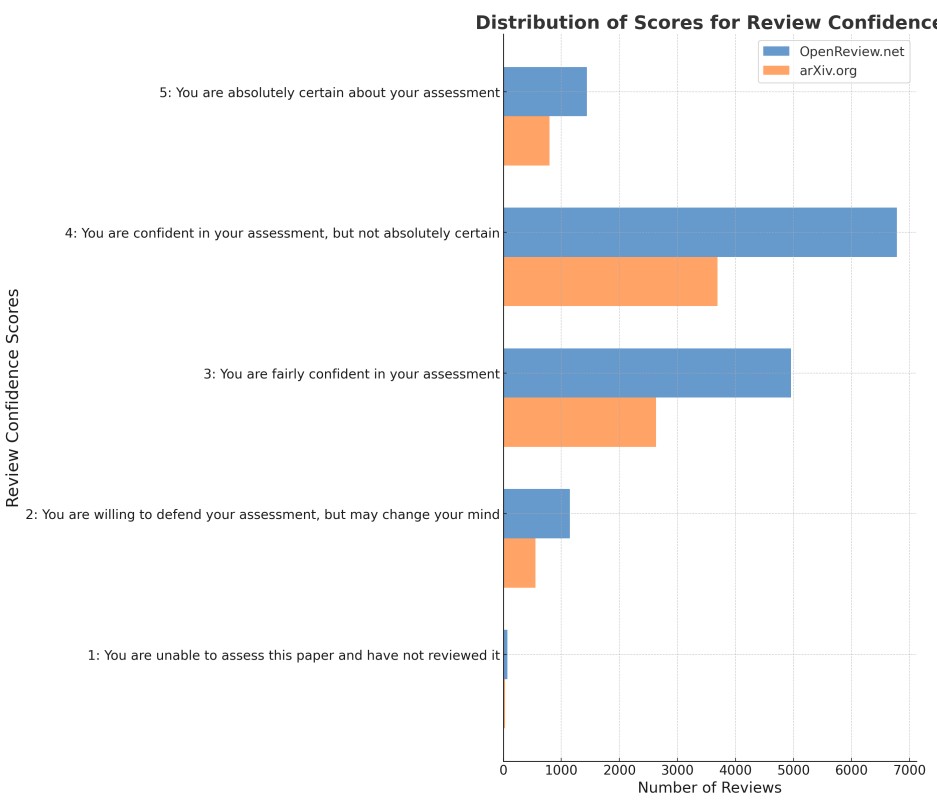

Figure 14: Distribution of human review confidence scores for papers publicly available on OpenReview.net and arXiv.org: Ranges are from 1 to 5, with 5 indicating absolute certainty in the assessment and 1 indicating an inability to assess the paper. A large number of reviewers were confident in their assessments, but not absolutely certain.

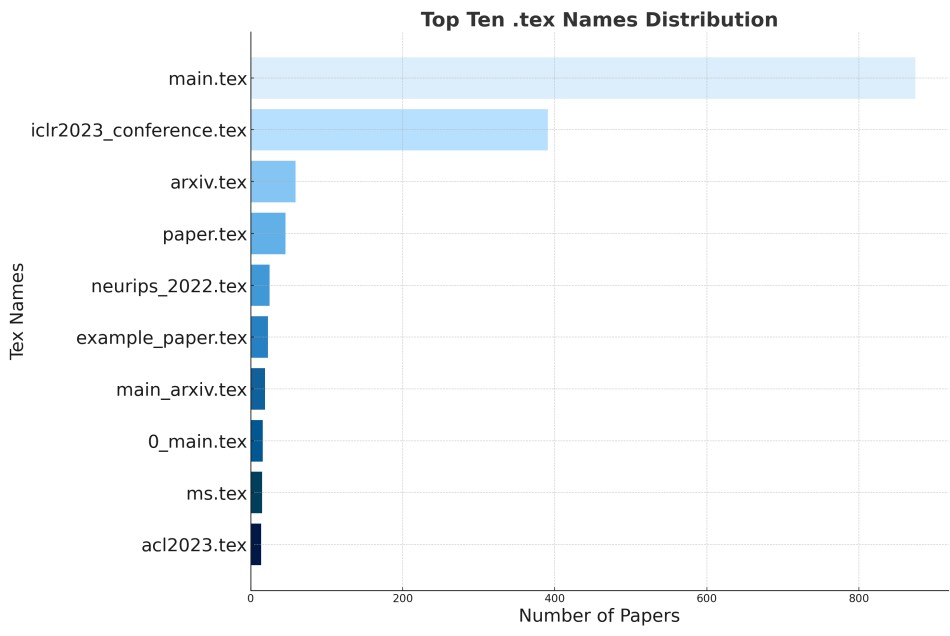

Figure 15: Distribution of the top ten .tex file names found for papers on arXiv.org

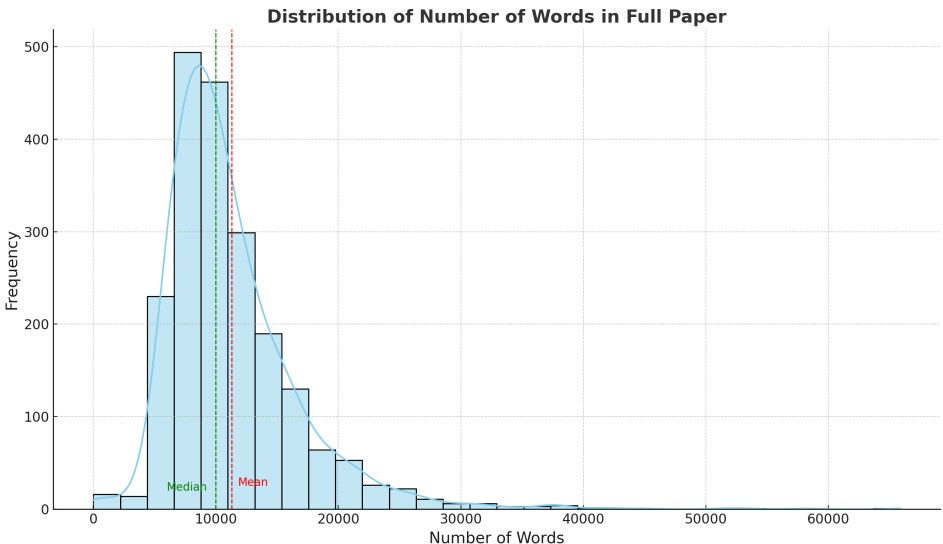

Figure 16: Distribution of the number of words in the full papers on arXiv.org. The mean is 11,298 words and the median 9,995 words.

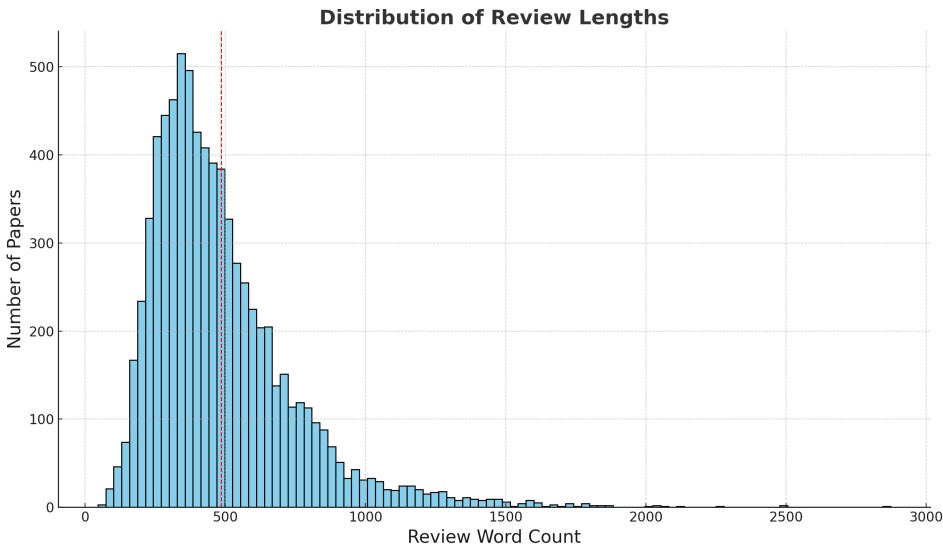

Figure 17: Distribution of human review lengths in number of words. The mean is 495 words.

## B  LLM REVIEW EXAMPLES

Tables 6, 7, 8, 12, 13, 14 show the ICLR review form used to prompt GPT-4 and example GPT-4 answers for both accepted and rejected papers. Tables 9, 10, 11, 15, 16, 17 show example human reviewer answers for both accepted and rejected papers.

| Category | Prompt and P5 Answer |
|---|---|
| Summary | **Prompt**: Summarize what the paper claims to contribute. Be positive and constructive.
**P5 Answer**:
The paper introduces GeneFace, a high-fidelity and generalizable approach to synthesize 3D talking faces from audio inputs using a Nerf-based method. The proposed framework aims to enhance methods used in film-making and virtual reality, addressing limitations of previous approaches that struggle with out-of-domain audio due to limited training data.
GeneFace is composed of three stages: an audio-to-motion converter, a motion domain adaptation stage, and finally, a motion-to-image model. The audio-to-motion stage employs a variational motion generator that learns from a large lip-reading corpus, handling the weak generalizability problem. The system then focuses on the domain adaptation stage, introducing an adversarial training pipeline that refines predictions, overcoming the gap between the generated landmarks and the target person's distribution. The final stage is the motion-to-image model, which uses a Nerf-based renderer to generate high-fidelity images.
The primary contributions of this paper are a three-stage framework improving generalizability to out-of-domain audio; a variational motion generator addressing the "mean face" problem and generating accurate facial landmarks, and experiments showing that GeneFace outperforms other state-of-the-art GAN-based and NeRF-based methods. |
| Correctness | **Prompt**: Please classify the paper on the following scale to indicate the correctness of the technical claims, experimental and research methodology and on whether the central claims of the paper are adequately supported with evidence. 4: All of the claims and statements are well-supported and correct. 3: Some of the paper's claims have minor issues. A few statements are not well-supported, or require small changes to be made correct. 2: Several of the paper's claims are incorrect or not well-supported. 1: The main claims of the paper are incorrect or not at all supported by theory or empirical results.
**P5 Answer**: 3 |
| Technical Novelty and Significance | **Prompt**: Please classify the paper on the following scale to indicate the technical novelty and significance of this paper. 4: The contributions are significant, and do not exist in prior works. 3: The contributions are significant and somewhat new. Aspects of the contributions exist in prior work. 2: The contributions are only marginally significant or novel. 1: The contributions are neither significant nor novel.
**P5 Answer**: 4 |
| Empirical Novelty and Significance | **Prompt**: Please classify the paper on the following scale to indicate the empirical novelty and significance of this paper. 4: The contributions are significant, and do not exist in prior works. 3: The contributions are significant and somewhat new. Aspects of the contributions exist in prior work. 2: The contributions are only marginally significant or novel. 1: The contributions are neither significant nor novel. NA: Not applicable.
**P5 Answer**: 3 |

Table 6: ICLR 2023 accepted paper evaluation prompts and example of P5 answers: Summary, correctness, technical novelty and significance, and empirical novelty and significance.

| Category | Prompt and P5 Answer |
|---|---|
| Strengths/Weaknesses | **Prompt**: List strong and weak points of the paper. Be as comprehensive as possible.
**P5 Answer**:
Strong Points: 1. The approach of the GeneFace system is novel and innovative. It addresses a current issue in the field of audio-to-visual synthesis in a fresh and unique way.
2. The three-stage framework proposed in the paper offers significant improvements to overcome the generalizability problems faced by current approaches for out-of-domain audio. This indicates a thoughtful understanding of the problem space.
3. The use of a Nerf-based renderer with a variational motion generator to prevent the "mean face" problem often encountered in similar systems is commendable. This shows an adept application of recent advances in AI to solve complex challenges in creative ways.
4. The extensive series of experiments conducted comparing GeneFace with other current state-of-the-art GAN-based and NeRF-based methods substantiates the claims made in the paper regarding the superior performance of their approach.
Weak Points: 1. The paper doesn't explicitly discuss the limitation of their training dataset. Information about its size, diversity, and domain-specific aspects could have been provided for a better understanding.
2. The proposed adversarial training pipeline for refining predictions in the motion domain adaptation stage lacks a detailed exposition. Without a clear description of the adversarial training pipeline, it becomes challenging to appreciate its potential impact fully.
3. While the paper provides comprehensive information about the GeneFace system, more visual representations or graphical models would have made the three-stage framework clearer and more accessible to the readers.
4. The paper does not discuss potential ethical concerns of such technology in depth. Given that GeneFace can be used to create very convincing fake videos (deepfakes), an in-depth discussion on potential misuse and mitigation strategies is warranted.
Overall, while the paper showcases a promising advancement in the field of audio-to-visual synthesis with the three-stage GeneFace system, it falls somewhat short in discussing its limitations, potential misuse, and provides limited explanation for the adversarial training pipeline used in the motion domain adaptation stage. I would recommend the authors to address these issues in a revised version of the paper. |
| Clarity, Quality, Novelty, and Reproducibility | **Prompt**: Please evaluate the clarity, quality, novelty, and reproducibility of the paper.
**P5 Answer**: Clarity: The paper is clearly organized into different sections, allowing readers to understand the progress from problem identification to solution development. The introduction and conclusion concisely summarize the main points and the proposed contributions. However, there are some complex discussions, particularly the adversarial training pipeline that could have been elucidated with more clarity. Visual aids like diagrams, charts, or schematics showing the structure of the model would have boosted the clarity score. Accordingly, the clarity score is 7/10.
Quality: The paper exhibits high standards in terms of scientific intellectual contribution. It demonstrates a deep understanding of the field and its challenges, proposes a novel mechanism to address these challenges, and supports the proposal with comprehensive experiments. However, it falters slightly in terms of explaining data limitations or detailing the adversarial training phase sufficiently. The quality score is 7.5/10.
Novelty: The novelty score of this paper is high. The proposed three-stage approach to synthesizing 3D talking faces from audio inputs using a Nerf-based renderer and a variational motion generator is both innovative and novel. The authors have applied recent advances in artificial intelligence to address a long-standing issue in audio-visual synthesis. Therefore, the novelty score is 8.5/10.
Reproducibility: Although the paper provides some technical detail on the GeneFace system, it does not provide sufficient specifics or disclose open-source code that would ensure reproducibility. In term of reproducibility, the paper scores 6.5/10.
Overall Assessment: The primary strength of the paper lies in the novelty of the proposed GeneFace system and the three-stage framework to synthesize 3D talking faces from audio inputs. However, the paper could considerably improve in terms of clarity, particularly by offering more detailed explanations and adding visual aids. Additionally, more information and discussion around potential misuse, data limitations, and reproducibility would make the paper more balanced. Thus, combining all scores, the overall score is 7.4/10.
Recommendation: The paper is a strong submission for ICLR 2023, showing creativity and a significant step in the field of audio-visual synthesis. However, I recommend acceptance on the condition that the authors to address observed shortcomings in terms of clarity, details about the adversarial training pipeline, data limitations, potential misuse, and reproducibility. |

Table 7: ICLR 2023 accepted paper evaluation prompts and example of P5 answers: Strengths/weaknesses, clarity, quality, novelty, and reproducibility

| Category | Prompt and P5 Answer |
|---|---|
| Summary of the Review | **Prompt**: Clearly state your initial recommendation (accept or reject) with one or two key reasons for this choice. Provide supporting arguments for your recommendation. 
 **P5 Answer**: 
 Initial Recommendation: Accept 
 Key Reasons: The submission presents a novel and innovative approach to synthesize 3D talking faces from audio signals using a NeRF-based renderer and a variational motion generator. Besides, the approach demonstrated superior performance compared to state-of-the-art GAN-based and NeRF-based methods. 
 Supporting Arguments: 
 1. Novelty: The three-stage GeneFace system for synthesizing 3D talking faces from audio inputs shows a new perspective in the field of audio-visual synthesis that could open avenues for further research. 
 2. Quality: Despite some weaknesses, the overall scientific quality of the paper is high. The methodology is well-designed, and the experiments conducted are comprehensive in supporting the claims made by the authors. 
 3. Potential Impact: Given the results shown, this work has the potential to significantly impact the field of audio-visual synthesis and applications such as film-making and virtual reality. 
 4. Improvements: While the paper lacks in certain areas like a detailed explanation of the adversarial training pipeline, limitations of training data, ethical implications, and reproducibility, these weaknesses are not insurmountable and could be addressed in a revised version. 
 Overall, the novelty and significant potential impact of the paper, along with the thoughtful three-stage design of the GeneFace system, outweigh the weaknesses. Therefore, I recommend accepting this paper for ICLR 2023, with the requirement that the authors address the highlighted issues in a revised version. |
| Flag for Ethics Review | **Prompt**: If there are no ethical issues with this paper please respond with NO. Otherwise if there are ethical issues with this paper please flag the paper for an ethics review by one of the following: Yes, Potentially harmful insights, methodologies and applications Yes, Legal compliance (e.g., GDPR, copyright, terms of use) Yes, Privacy, security and safety, Yes, Potentially harmful insights, methodologies and applications Yes, Discrimination / bias / fairness concerns Yes, Responsible research practice (e.g., human subjects, data release) Yes, Research integrity issues (e.g., plagiarism, dual submission) Yes, Other reasons (please specify below) 
 **P5 Answer**: Yes, Potentially harmful insights, methodologies and applications |
| Overall Score | **Prompt**: Please classify the paper on the following scale and provide an "overall score" for this submission. Provide your output with the numerical rating. You are encouraged to use the full range of scores. 10: strong accept, should be highlighted at the conference 8: accept, good paper 6: marginally above the acceptance threshold 5: marginally below the acceptance threshold 3: reject, not good enough 1: strong reject 
 **P5 Answer**: 7 |
| Confidence | **Prompt**: Please classify the paper on the following scale to indicate how confident you are in your review. Provide your output with the numerical rating. 5: You are absolutely certain about your assessment. You are very familiar with the related work and checked the math/other details carefully. 4: You are confident in your assessment, but not absolutely certain. It is unlikely, but not impossible, that you did not understand some parts of the submission or that you are unfamiliar with some pieces of related work. 3: You are fairly confident in your assessment. It is possible that you did not understand some parts of the submission or that you are unfamiliar with some pieces of related work. Math/other details were not carefully checked. 2: You are willing to defend your assessment, but it is quite likely that you did not understand the central parts of the submission or that you are unfamiliar with some pieces of related work. Math/other details were not carefully checked. 1: You are unable to assess this paper and have alerted the ACs to seek an opinion from different reviewers. 
 **P5 Answer**: 4 |

Table 8: ICLR 2023 accepted paper evaluation prompts and example of P5 answers: Summary of the review, flag for ethics review, overall score, and confidence

| Category | Prompt and Human Answer |
|---|---|
| Summary | **Prompt**: Summarize what the paper claims to contribute. Be positive and constructive.
**Human Answer**:
This paper proposed an end-to-end NeRF-based method, for talking face generation. It was trying to solve 2 problems: 1) weak generalizability due to small scale of training data, 2) "mean face" result: bad image quality and bad lip-synchronization. This paper proposed 3 parts: 1) Variational Motion Generator used to generate landmarks from audio, 2) Domain Adaptative Post-Net for landmark refinement, and 3) NeRF based Renderer for final frame generation. It produces good talking face generation results. |
| Correctness | **Prompt**: Please classify the paper on the following scale to indicate the correctness of the technical claims, experimental and research methodology and on whether the central claims of the paper are adequately supported with evidence. 4: All of the claims and statements are well-supported and correct. 3: Some of the paper's claims have minor issues. A few statements are not well-supported, or require small changes to be made correct. 2: Several of the paper's claims are incorrect or not well-supported. 1: The main claims of the paper are incorrect or not at all supported by theory or empirical results.
**Human Answer**: 4 |
| Technical Novelty and Significance | **Prompt**: Please classify the paper on the following scale to indicate the technical novelty and significance of this paper. 4: The contributions are significant, and do not exist in prior works. 3: The contributions are significant and somewhat new. Aspects of the contributions exist in prior work. 2: The contributions are only marginally significant or novel. 1: The contributions are neither significant nor novel.
**Human Answer**: 3 |
| Empirical Novelty and Significance | **Prompt**: Please classify the paper on the following scale to indicate the empirical novelty and significance of this paper. 4: The contributions are significant, and do not exist in prior works. 3: The contributions are significant and somewhat new. Aspects of the contributions exist in prior work. 2: The contributions are only marginally significant or novel. 1: The contributions are neither significant nor novel. NA: Not applicable.
**Human Answer**: 3 |

Table 9: ICLR 2023 accepted paper evaluation prompts and example of human reviewer answers: Summary, correctness, technical novelty and significance, and empirical novelty and significance.

| Category | Prompt and Human Answer |
|---|---|
| Strengths/Weaknesses | **Prompt**: List strong and weak points of the paper. Be as comprehensive as possible.
**Human Answer**:
Strength: This paper has well-organized structure, and it is clear in logicality. This work has well-designed network structure and proposed structure, which can actually solve proposed problems. This work combines previous works and gives an new approach to generating talking face in another way, which moves the NeRF-based method field forward a bit.
Weaknesses: Some evaluation metric do not outperform other methods, this work could design more evaluation metrics to evaluate the results. The generated face in the demo video is moving around, maybe consider temporal information and design into network structure. The lip-synchronization is not as good as wav2lip, the synchronization can be improved. |
| Clarity, Quality, Novelty, and Reproducibility | **Prompt**: Please evaluate the clarity, quality, novelty, and reproducibility of the paper.
**Human Answer**: This paper has well-organized structure, and it is clear in logicality. This work proposed three sub-network and they can solve problems. It is reproducible work according to the designed network structure and experiment results. |

Table 10: ICLR 2023 accepted paper evaluation prompts and example of human reviewer answers: Strengths/weaknesses, clarity, quality, novelty, and reproducibility

| Category | Prompt and Human Answer |
|---|---|
| Summary of the Review | **Prompt**: Clearly state your initial recommendation (accept or reject) with one or two key reasons for this choice. Provide supporting arguments for your recommendation. 
 **Human Answer**: 
 The paper is overall well written and much of it is well described. It combines previous works and proposes an new approach to generating 3D talking face. It produces better results than previous NeRF-based work. I recommend this work. |
| Flag for Ethics Review | **Prompt**: If there are no ethical issues with this paper please respond with NO. Otherwise if there are ethical issues with this paper please flag the paper for an ethics review by one of the following: Yes, Potentially harmful insights, methodologies and applications Yes, Legal compliance (e.g., GDPR, copyright, terms of use) Yes, Privacy, security and safety, Yes, Potentially harmful insights, methodologies and applications Yes, Discrimination / bias / fairness concerns Yes, Responsible research practice (e.g., human subjects, data release) Yes, Research integrity issues (e.g., plagiarism, dual submission) Yes, Other reasons (please specify below) 
 **Human Answer**: Yes, Privacy, security and safety |
| Overall Score | **Prompt**: Please classify the paper on the following scale and provide an "overall score" for this submission. Provide your output with the numerical rating. You are encouraged to use the full range of scores. 10: strong accept, should be highlighted at the conference 8: accept, good paper 6: marginally above the acceptance threshold 5: marginally below the acceptance threshold 3: reject, not good enough 1: strong reject 
 **Human Answer**: 6 |
| Confidence | **Prompt**: Please classify the paper on the following scale to indicate how confident you are in your review. Provide your output with the numerical rating. 5: You are absolutely certain about your assessment. You are very familiar with the related work and checked the math/other details carefully. 4: You are confident in your assessment, but not absolutely certain. It is unlikely, but not impossible, that you did not understand some parts of the submission or that you are unfamiliar with some pieces of related work. 3: You are fairly confident in your assessment. It is possible that you did not understand some parts of the submission or that you are unfamiliar with some pieces of related work. Math/other details were not carefully checked. 2: You are willing to defend your assessment, but it is quite likely that you did not understand the central parts of the submission or that you are unfamiliar with some pieces of related work. Math/other details were not carefully checked. 1: You are unable to assess this paper and have alerted the ACs to seek an opinion from different reviewers. 
 **Human Answer**: 3 |

Table 11: ICLR 2023 accepted paper evaluation prompts and example of human reviewer answers: Summary of the review, flag for ethics review, overall score, and confidence

| Category | Prompt and P5 Answer |
|---|---|
| Summary | **Prompt**: Summarize what the paper claims to contribute. Be positive and constructive.
**P5 Answer**:
The paper aims to tackle the issue of increased computational expense in calculating the batch gradient in each iteration when using the Iterative Hard Thresholding (IHT) algorithm for solving sparse optimizations. This becomes problematic in cases of high-dimensional data or machine learning applications. The authors propose a variant of the Stochastic IHT algorithm that uses a mini-batch gradient to remedy the computational expensive process.
The authors claim a few main contributions: 1. They establish a novel sparse stochastic gradient descent property of the Hard Thresholding (HT) operator. 2. For a given step size, they find a lower bound on the size of the mini-batch that ensures the expected descent of the objective value function after hard-thresholding. 3. They show that the sequence generated by the mini-batch SIHT algorithm is supermartingale and converges with a probability of one. 4. They demonstrate that for certain classes of functions in problem (P), the sum of norm squared of individual gradients, evaluated at every point, is proportionate to the norm of the batch gradient, where the proportionality only depends on the data.
However, some key concerns remain. All these results are claimed without explicitly illustrating any significant experimental or practical scaling trends. Furthermore, the work lacks any comparative benchmarks against recent methods. Also, some dense theory presentation makes the paper less accessible to a broader readership. Useful connections to machine learning algorithms, networks, and commonly faced issues are not convincingly made with numerous concepts assumed to be known by readers without enough explanation. Finally, potential limitations or downsides of the authors' approach remain unaddressed. Therefore, from a critic's point of view, this work could score around 4/10. Some recommendations would be to simplify complex theoretical presentations, provide comparative benchmarks, and explicitly discuss limitations for balance and context. |
| Correctness | **Prompt**: Please classify the paper on the following scale to indicate the correctness of the technical claims, experimental and research methodology and on whether the central claims of the paper are adequately supported with evidence. 4: All of the claims and statements are well-supported and correct. 3: Some of the paper's claims have minor issues. A few statements are not well-supported, or require small changes to be made correct. 2: Several of the paper's claims are incorrect or not well-supported. 1: The main claims of the paper are incorrect or not at all supported by theory or empirical results.
**P5 Answer**: 2 |
| Technical Novelty and Significance | **Prompt**: Please classify the paper on the following scale to indicate the technical novelty and significance of this paper. 4: The contributions are significant, and do not exist in prior works. 3: The contributions are significant and somewhat new. Aspects of the contributions exist in prior work. 2: The contributions are only marginally significant or novel. 1: The contributions are neither significant nor novel.
**P5 Answer**: 2 |
| Empirical Novelty and Significance | **Prompt**: Please classify the paper on the following scale to indicate the empirical novelty and significance of this paper. 4: The contributions are significant, and do not exist in prior works. 3: The contributions are significant and somewhat new. Aspects of the contributions exist in prior work. 2: The contributions are only marginally significant or novel. 1: The contributions are neither significant nor novel. NA: Not applicable.
**P5 Answer**: 2 |

Table 12: ICLR 2023 rejected paper evaluation prompts and example of P5 answers: Summary, correctness, technical novelty and significance, and empirical novelty and significance.

| Category | Prompt and P5 Answer |
|---|---|
| Strengths/Weaknesses | **Prompt**: List strong and weak points of the paper. Be as comprehensive as possible. 
 **P5 Answer**: 
 Strong Points: 1. The paper addresses a significant challenge in applying IHT for sparse optimizations in high-dimensional data or machine learning applications. 2. The authors propose an innovative variant of the Stochastic IHT algorithm employing a mini-batch gradient which is computationally cheaper. 3. The establishment of a novel sparse stochastic gradient descent property of the HT operator is a potentially significant contribution. 4. The authors determine the minimum mini-batch size based on a given step size, permitting expected descent of the objective value function after hard-thresholding. 5. The proof that the sequence generated by the mini-batch SIHT algorithm converges with a high probability and behaves like a supermartingale could provide strong theoretical underpinning. 
 Weak Points: 1. The lack of significant experimental or practical scaling trends limits the applicability and demonstrability of the proposed methods. 2. Absence of comparison benchmarks against other current methods weakens the paper's argument for its proposed methodology. 3. The prolonged sections of dense theory could alienate readers who may not be highly familiar with the content, thereby reducing accessibility. 4. There is insufficient contextualization between numerous concepts and their corresponding relevance to machine learning algorithms and networks. 5. The work does not address any of its potential limitations or downsides, limiting its observed objectivity and balance. 
 Given these reasons, the authors ought to strive for improved engagement with the readership by simplifying theoretical presentations, conducting and presenting experimental or practical proofs, providing comparative benchmarks, and discussing limitations explicitly for a well-rounded view. |
| Clarity, Quality, Novelty, and Reproducibility | **Prompt**: Please evaluate the clarity, quality, novelty, and reproducibility of the paper. 
 **P5 Answer**: Clarity: The paper is somewhat hard to follow for readers not already very familiar with the specific concepts referenced. While it is organized logically, a significant portion of the content is mired in complex, dense theory that may make it inaccessible to a broader readership. Illustrations, intuitive descriptions, and more contextualizations should be included to clarify the derived theorems and properties. The authors may also want to invest in building a more convincing narrative regarding the problems they're addressing and the potential of their proposed solutions in actual applications. 
 Quality: Research quality appears to be reasonably strong with stated claims backed up by mathematical formulations and reasoning. However, the paper does not provide any implementation details or extensive insights from experiments, testing, or simulations, which can buttress the claims. The applicability of the theoretical findings to practical scenarios is thus not firmly demonstrated, affecting the perceived quality of the paper. Furthermore, the work does not engage with other related research in a comparative way, which could have validated the uniqueness and potency of their offerings against existing methods. 
 Novelty: The novel components in this paper include a new sparse stochastic gradient descent property of the HT operator and the establishment of a lower bound on mini-batch size for the mini-batch SIHT algorithm. To academic specialists, these findings could represent some healthy progress. However, with the paper lacking an experimental component or concrete prototypes to validate these novelty points, their actual effectiveness, utility, or applicability could be unclear to many readers. 
 Reproducibility: The authors focus the paper on theory and do not report on any empirical investigations they may have carried out. Although the theorems and proofs are presented, details that would help to reproduce these findings such as data sets, code snippets, or certain specifics of the experimental setup (if one was used), are missing in the paper. As a result, the reproducibility of this work is currently low. 
 In accordance to the provided guidelines, my suggested scores would be: Clarity: 4, Quality: 5, Novelty: 6, Reproducibility: 3. The authors need to work on improving the presentation, comparative benchmarks, offering experimental / practical details, and discussing limitations as part of enhancing the overall appeal and substance of this work. |

Table 13: ICLR 2023 rejected paper evaluation prompts and example of P5 answers: Strengths/weaknesses, clarity, quality, novelty, and reproducibility

| Category | Prompt and P5 Answer |
|---|---|
| Summary of the Review | **Prompt**: Clearly state your initial recommendation (accept or reject) with one or two key reasons for this choice. Provide supporting arguments for your recommendation.
**P5 Answer**:
Initial Recommendation: Reject
Reasons: 1. Lack of Experimental Verification: The paper lacks significant practical experimental verification. Although theoretical analysis is important, practical demonstrations are also necessary to prove the practical feasibility of the approach proposed.
2. Absence of Comparative Benchmarks: The paper doesn't justify the novel contributions by including comparative benchmarks against existing approaches. This greatly undermines the demonstrated efficiency of the proposed method.
Supporting Arguments:
1. While the paper provides some motivated theoretical insights, their practicality and efficacy remain untested. To ensure credibility, the authors need to complement their theory with experimental evidence illustrating the actionable significance of their assays.
2. It's unclear how the proposed approach fares against existing solutions since there isn't any comparative benchmark provided in the paper.
3. The dense theory presentation of content and lack of an accessible exposition can make it harder for readers not well-versed with the concepts to appreciate the innovations presented. Therefore, making these sections more accessible can significantly improve the appeal and engagement of the paper,
4. Lastly, the authors fail to discuss the potential shortcomings of their approach which can make the work seem unbalanced and poorly contextualized. Thus, acknowledging and outlining limitations can serve to enhance the transparency and credibility of their findings.
While the paper's methodology exhibits novelty, the lack of experimental groundwork and comparative benchmarks limit its viability for presentation in its current form. These expansive changes can result in a notable lift in the paper's bigger capabilities, hence, warranting adjustments to be made before another review attempt. |
| Flag for Ethics Review | **Prompt**: If there are no ethical issues with this paper please respond with NO. Otherwise if there are ethical issues with this paper please flag the paper for an ethics review by one of the following: Yes, Potentially harmful insights, methodologies and applications Yes, Legal compliance (e.g., GDPR, copyright, terms of use) Yes, Privacy, security and safety, Yes, Potentially harmful insights, methodologies and applications Yes, Discrimination / bias / fairness concerns Yes, Responsible research practice (e.g., human subjects, data release) Yes, Research integrity issues (e.g., plagiarism, dual submission) Yes, Other reasons (please specify below)
**P5 Answer**: NO |
| Overall Score | **Prompt**: Please classify the paper on the following scale and provide an "overall score" for this submission. Provide your output with the numerical rating. You are encouraged to use the full range of scores. 10: strong accept, should be highlighted at the conference 8: accept, good paper 6: marginally above the acceptance threshold 5: marginally below the acceptance threshold 3: reject, not good enough 1: strong reject
**P5 Answer**: 3 |
| Confidence | **Prompt**: Please classify the paper on the following scale to indicate how confident you are in your review. Provide your output with the numerical rating. 5: You are absolutely certain about your assessment. You are very familiar with the related work and checked the math/other details carefully. 4: You are confident in your assessment, but not absolutely certain. It is unlikely, but not impossible, that you did not understand some parts of the submission or that you are unfamiliar with some pieces of related work. 3: You are fairly confident in your assessment. It is possible that you did not understand some parts of the submission or that you are unfamiliar with some pieces of related work. Math/other details were not carefully checked. 2: You are willing to defend your assessment, but it is quite likely that you did not understand the central parts of the submission or that you are unfamiliar with some pieces of related work. Math/other details were not carefully checked. 1: You are unable to assess this paper and have alerted the ACs to seek an opinion from different reviewers.
**P5 Answer**: 4 |

Table 14: ICLR 2023 rejected paper evaluation prompts and example of P5 answers: Summary of the review, flag for ethics review, overall score, and confidence

| Category | Prompt and Human Answer |
|---|---|
| Summary | **Prompt**: Summarize what the paper claims to contribute. Be positive and constructive. |
| | **Human Answer**: |
| | The authors study mini-batch stochastic iterative hard thresholding (IHT). IHT is a well known algorithm for sparse optimization with extensive literature. IHT is essentially $\ell_2$-projected gradient descent on the set of s-sparse vectors. Given the ML-wide focus on stochastic optimization algorithms, it is natural to study the guarantees of stochastic IHT. |
| | The main result of the paper is a martingale convergence result that claims stochastic IHT converges to the optimal function value, although no convergence rate is given. |
| Correctness | **Prompt**: Please classify the paper on the following scale to indicate the correctness of the technical claims, experimental and research methodology and on whether the central claims of the paper are adequately supported with evidence. 4: All of the claims and statements are well-supported and correct. 3: Some of the paper's claims have minor issues. A few statements are not well-supported, or require small changes to be made correct. 2: Several of the paper's claims are incorrect or not well-supported. 1: The main claims of the paper are incorrect or not at all supported by theory or empirical results. |
| | **Human Answer**: 2 |
| Technical Novelty and Significance | **Prompt**: Please classify the paper on the following scale to indicate the technical novelty and significance of this paper. 4: The contributions are significant, and do not exist in prior works. 3: The contributions are significant and somewhat new. Aspects of the contributions exist in prior work. 2: The contributions are only marginally significant or novel. 1: The contributions are neither significant nor novel. |
| | **Human Answer**: 2 |
| Empirical Novelty and Significance | **Prompt**: Please classify the paper on the following scale to indicate the empirical novelty and significance of this paper. 4: The contributions are significant, and do not exist in prior works. 3: The contributions are significant and somewhat new. Aspects of the contributions exist in prior work. 2: The contributions are only marginally significant or novel. 1: The contributions are neither significant nor novel. NA: Not applicable. |
| | **Human Answer**: Not applicable |

Table 15: ICLR 2023 rejected paper evaluation prompts and example of human reviewer answers: Summary, correctness, technical novelty and significance, and empirical novelty and significance.

| Category | Prompt and Human Answer |
|---|---|
| Strengths/Weaknesses | **Prompt**: List strong and weak points of the paper. Be as comprehensive as possible. |
| | **Human Answer**: |
| | Strengths - Advancing the theoretical understanding of stochastic IHT is an important task. - The writing is generally easy to follow. |
| | Weaknesses - The literature review is quite limited. - The proofs are missing, so it is not possible to fully verify the claimed results. - In Claim 2, the minimum singular value should be 0, because the matrix is low-rank. If the authors define minimum singular value differently, it is hard to know because the proof is not included. - In Claim 2, an upper bound is derived for c (10). Based on this, $c$ seems to be $\geq N^2$. When I plug that back into Theorem 2, I get a batch size of roughly $N - \Theta(1)$. This does not seem like a great choice for a batch size, which typically should be much less than $N$. It would be great if the authors could comment on whether I am missing something here. - There are no experiments, which could shed light for example on what is the value of c in practice. |
| Clarity, Quality, Novelty, and Reproducibility | **Prompt**: Please evaluate the clarity, quality, novelty, and reproducibility of the paper. |
| | **Human Answer**: On novelty, I find that at least one very related paper is missing from the discussion [1] (probably others too). This paper presents a convergence analysis of stochastic IHT for functions that have the restricted smooth and strongly convexity assumptions. By the way, the strong convexity assumption can be guaranteed by regularizing the loss function. |
| | On reproducibility, some of the results like Lemmas 3-4 seem true, while e.g. Claim 2 is hard to verify. Again, since the proofs aren't included, this has a negative effect on reproducibility. |
| | [1] Ac/dc: Alternating compressed/decompressed training of deep neural networks Peste, Alexandra and Iofinova, Eugenia and Vladu, Adrian and Alistarh, Dan |

Table 16: ICLR 2023 rejected paper evaluation prompts and example of human reviewer answers: Strengths/weaknesses, clarity, quality, novelty, and reproducibility

| Category | Prompt and Human Answer |
|---|---|
| Summary of the Review | **Prompt**: Clearly state your initial recommendation (accept or reject) with one or two key reasons for this choice. Provide supporting arguments for your recommendation.
**Human Answer**:
In summary, while I like the problem of minibatch stochastic IHT and the approach, I find that there are significant issues with novelty that should be resolved in the literature review section, as well as my other concerns outlined above. I encourage the authors to continue working on the manuscript since I think this is an important subject. |
| Flag for Ethics Review | **Prompt**: If there are no ethical issues with this paper please respond with NO. Otherwise if there are ethical issues with this paper please flag the paper for an ethics review by one of the following: Yes, Potentially harmful insights, methodologies and applications Yes, Legal compliance (e.g., GDPR, copyright, terms of use) Yes, Privacy, security and safety, Yes, Potentially harmful insights, methodologies and applications Yes, Discrimination / bias / fairness concerns Yes, Responsible research practice (e.g., human subjects, data release) Yes, Research integrity issues (e.g., plagiarism, dual submission) Yes, Other reasons (please specify below)
**Human Answer**: NO |
| Overall Score | **Prompt**: Please classify the paper on the following scale and provide an "overall score" for this submission. Provide your output with the numerical rating. You are encouraged to use the full range of scores. 10: strong accept, should be highlighted at the conference 8: accept, good paper 6: marginally above the acceptance threshold 5: marginally below the acceptance threshold 3: reject, not good enough 1: strong reject
**Human Answer**: 3 |
| Confidence | **Prompt**: Please classify the paper on the following scale to indicate how confident you are in your review. Provide your output with the numerical rating. 5: You are absolutely certain about your assessment. You are very familiar with the related work and checked the math/other details carefully. 4: You are confident in your assessment, but not absolutely certain. It is unlikely, but not impossible, that you did not understand some parts of the submission or that you are unfamiliar with some pieces of related work. 3: You are fairly confident in your assessment. It is possible that you did not understand some parts of the submission or that you are unfamiliar with some pieces of related work. Math/other details were not carefully checked. 2: You are willing to defend your assessment, but it is quite likely that you did not understand the central parts of the submission or that you are unfamiliar with some pieces of related work. Math/other details were not carefully checked. 1: You are unable to assess this paper and have alerted the ACs to seek an opinion from different reviewers.
**Human Answer**: 4 |

Table 17: ICLR 2023 rejected paper evaluation prompts and example of human reviewer answers: Summary of the review, flag for ethics review, overall score, and confidence

| Category | Prompt and P5 Answer |
|---|---|
| Summary of the Review | **Prompt**: Clearly state your initial recommendation (accept or reject) with one or two key reasons for this choice. Provide supporting arguments for your recommendation.
**P5 Answer**:
Initial Recommendation: Reject
Key Reasons for Rejection:
1. Lack of Evidence and Rigorous Validation: The authors state that they have designed efficient attack algorithms, but no substantial empirical evidence or case studies are provided to support these claims. The conspicuous absence of practical illustrations significantly weakens the presented arguments.
2. Theoretical Focus with Minimal Consideration for Real-World Applicability: While there are decent theoretical foundations and analytical rigor presented throughout the paper, it lacks a well-illustrated methodology or consideration for practical implications that could reaffirm the utility of the proposed work in real-world scenarios.
Supporting Arguments:
1. The paper presents an interesting perspective regarding security threats to adversarial bandits, a niche that is indeed intriguing. However, the presentation substantially lacks evidence-based arguments and ends up being too theoretical in most parts. Any claims made require substantiation through rigorous empirical analysis, which the paper seems to inadequately provide.
2. The theoretical designs discussed would provide grounds for further exploration but require resemblance and alignment to practical scenarios. This alignment is lacking and handicaps the strength of this submission in efficiently exhibiting the relevance and potential of proposed models.
3. Overall, the research demonstrates potential but is unable to effectively convince the reader about its practical and impactful implications. Lack of strong experimental support, complexity in interpreting the results presented, and insufficient acknowledgment of the broader range of existing work dilute the comprehension and appreciation of the deliverance.
4. Key messages deriving the novelty of the work get lost due to not-so-well-illustrated background study, unclear results, and inadequately examined novelty. Upon accommodating revisions targeting these concerns, the research could turn out to be a qualified proposal exhibiting more effective and clarified novelties.
Resolving these issues and incorporating substantial empirical validation will significantly augment the concreteness, relevance, and palpability of the work, potentially setting requisites for its acceptance. |
| Overall Score | **Prompt**: Please classify the paper on the following scale and provide an "overall score" for this submission. Provide your output with the numerical rating. You are encouraged to use the full range of scores. 10: strong accept, should be highlighted at the conference 8: accept, good paper 6: marginally above the acceptance threshold 5: marginally below the acceptance threshold 3: reject, not good enough 1: strong reject
**P5 Answer**: 4 |
| Confidence | **Prompt**: Please classify the paper on the following scale to indicate how confident you are in your review. Provide your output with the numerical rating. 5: You are absolutely certain about your assessment. You are very familiar with the related work and checked the math/other details carefully. 4: You are confident in your assessment, but not absolutely certain. It is unlikely, but not impossible, that you did not understand some parts of the submission or that you are unfamiliar with some pieces of related work. 3: You are fairly confident in your assessment. It is possible that you did not understand some parts of the submission or that you are unfamiliar with some pieces of related work. Math/other details were not carefully checked. 2: You are willing to defend your assessment, but it is quite likely that you did not understand the central parts of the submission or that you are unfamiliar with some pieces of related work. Math/other details were not carefully checked. 1: You are unable to assess this paper and have alerted the ACs to seek an opinion from different reviewers.
**P5 Answer**: 4 |

Table 18: Example of P5 review after adding metric errors to the paper: Summary of the review, overall score and confidence.

| Category | Prompt and P5 Answer |
|---|---|
| Summary of the Review | **Prompt**: Clearly state your initial recommendation (accept or reject) with one or two key reasons for this choice. Provide supporting arguments for your recommendation.
**P5 Answer**:
Initial Recommendation: Reject
Key Reasons:
1. Insufficient Theoretical Justifications: The paper fails to strategically present sufficient theoretical motivation for its proposed approach that might justify its relevance over other existing methods, thereby putting the novelty and originality of the paper into question.
2. Limited Experimentations: The quality of the paper is hampered significantly by the fact that it only evaluates its method on validation sets without considering any application on independent test sets for generalizability. The experimentation procedure noticeably lacking comparison with other state-of-the-art methods, making it difficult to consider the observed improvements as valid or significant.
Supporting Arguments:
1. Theoretical reasoning is a vital aspect that differentiates any research piece. Sufficient background literature and logical rationale must support the proposed approach. The lack of these seriously dents the credibility and thus pulls the overall score down for this submission.
2. The robustness analysis, potential challenges, and limitations of the proposed algorithm are starkly missing from the paper. Such discourse would offer insights into the practice readiness of the model and its fitness for real-world scenarios.
3. For efficacy and the subsequent impact of this research work on the academic community, a robust experimental methodology constituting benchmark comparisons and generalizability study would be crucial. Else, the research stands isolated from the context.
4. A clear reproducibility profile is a must for driving a healthy trend in academic contributions, as it allows for potential comparative studies, enhancements, and to corroborate the results. The absence of such essential details in this submission marks another strike against it.
Conclusively, my evaluation suggests that the paper holds some promise with its introduced method, the CIM, placing itself on a novel idea platform. However, substantial scope for improvement in theory justification, experimentation design, and the reproducible outlook before it might hold a ground in the ICLR 2023 niche persists. Therefore, at its current state, it does not qualify for meeting the year's acceptance threshold. |
| Overall Score | **Prompt**: Please classify the paper on the following scale and provide an "overall score" for this submission. Provide your output with the numerical rating. You are encouraged to use the full range of scores. 10: strong accept, should be highlighted at the conference 8: accept, good paper 6: marginally above the acceptance threshold 5: marginally below the acceptance threshold 3: reject, not good enough 1: strong reject
**P5 Answer**: 4 |
| Confidence | **Prompt**: Please classify the paper on the following scale to indicate how confident you are in your review. Provide your output with the numerical rating. 5: You are absolutely certain about your assessment. You are very familiar with the related work and checked the math/other details carefully. 4: You are confident in your assessment, but not absolutely certain. It is unlikely, but not impossible, that you did not understand some parts of the submission or that you are unfamiliar with some pieces of related work. 3: You are fairly confident in your assessment. It is possible that you did not understand some parts of the submission or that you are unfamiliar with some pieces of related work. Math/other details were not carefully checked. 2: You are willing to defend your assessment, but it is quite likely that you did not understand the central parts of the submission or that you are unfamiliar with some pieces of related work. Math/other details were not carefully checked. 1: You are unable to assess this paper and have alerted the ACs to seek an opinion from different reviewers.
**P5 Answer**: 2 |

Table 19: Example of P5 review after adding theoretical errors to the paper: Summary of the review, overall score and confidence.

| Category | Prompt and P5 Answer |
|---|---|
| Summary of the Review | **Prompt**: Clearly state your initial recommendation (accept or reject) with one or two key reasons for this choice. Provide supporting arguments for your recommendation.
**P5 Answer**:
Initial Recommendation: Reject
Reasons: Firstly, the paper is insufficient in providing comprehensive experimental validation and comparative studies with other models to back up the claims about the efficacy of their proposed technique. For example, it lacks a comparison with mechanisms such as the MIM or BEiT and doesn't offer enough evidence to support the superiority of their ResPix and RepDet objectives. Secondly, the heavy reliance on existing techniques reduces its novelty. However, the concepts brought out are promising and if better validated methodologically, could be worth acceptance.
Supporting Arguments: 1. Lack of Comparisons: For the novel aspect of Corrupted Image Modeling (CIM), the paper does not offer comparative studies with other techniques in the field. The verification of the superiority of the novel components, including the ResPix and RepDet objectives, is also not well proven against existing pre-training objectives.
2. Dependence on known approaches: While it's not a flaw to build upon existing theories or methods, the paper appears to lean heavily on established techniques like BEiT and the Transformer family. Thereby, limiting its own novelty.
3. Insufficient validation: The experimental setup presented doesn't demonstrate conclusively that the proposed approach is efficient or better than others. Providing entire process metrics against familiar methods can strengthen this aspect.
4. Clarity and Reproducibility Issues: The complex concepts and theoretical explanations aren't properly illustrated. Essential details are missing which can lead to obstacles in understanding and reproducing the results.
Overall, despite its novel proposals, the shortcomings regarding validation, comparison and dependency on familiar methods warrant a rejection at the current review. The paper needs extensive updates and improvements to receive a higher score. |
| Overall Score | **Prompt**: Please classify the paper on the following scale and provide an "overall score" for this submission. Provide your output with the numerical rating. You are encouraged to use the full range of scores. 10: strong accept, should be highlighted at the conference 8: accept, good paper 6: marginally above the acceptance threshold 5: marginally below the acceptance threshold 3: reject, not good enough 1: strong reject
**P5 Answer**: 4 |
| Confidence | **Prompt**: Please classify the paper on the following scale to indicate how confident you are in your review. Provide your output with the numerical rating. 5: You are absolutely certain about your assessment. You are very familiar with the related work and checked the math/other details carefully. 4: You are confident in your assessment, but not absolutely certain. It is unlikely, but not impossible, that you did not understand some parts of the submission or that you are unfamiliar with some pieces of related work. 3: You are fairly confident in your assessment. It is possible that you did not understand some parts of the submission or that you are unfamiliar with some pieces of related work. Math/other details were not carefully checked. 2: You are willing to defend your assessment, but it is quite likely that you did not understand the central parts of the submission or that you are unfamiliar with some pieces of related work. Math/other details were not carefully checked. 1: You are unable to assess this paper and have alerted the ACs to seek an opinion from different reviewers.
**P5 Answer**: 4 |

Table 20: Example of P5 review after adding exaggerated claims in the paper: Summary of the review, overall score and confidence.

