# OpenReview forum: "OpenReviewer: Mitigating Challenges in LLM Reviewing"
_ICLR.cc/2024/Conference — ICLR 2024 Conference Withdrawn Submission_

### Official Review · Reviewer_koKB · 2023-10-24

**Soundness:** 2 fair
**Presentation:** 2 fair
**Contribution:** 1 poor
**Rating:** 3
**Confidence:** 4

**Summary:**

This paper explores the potential of large language models (LLMs) in reviewing research papers, presenting a proof-of-concept tool called OpenReviewer that can provide reviews by leveraging an LLM. The authors present an analysis of ICLR papers, comparing human and LLM ratings of these papers. They also explore the impact of a watermarking technique as well the use of reviewer, area chair, and other guidelines in the LLM prompts on the ratings. A blind human evaluation was conducted to compare ratings.

**Strengths:**

This is certainly a topic of current debate. LLMs are likely being used by reviewers to aid, so it's important for us to understand their limitations, as well as whether and when they can help humans.

**Weaknesses:**

My first concern is that, even though one may advocate for the use of LLMs throughout the review process, the human should always be in the loop. Even in this case, the various pitfalls of using LLMs should be discussed. This paper goes a step further and analyzes the use of LLMs for reviewing without humans, but lacks any meaningful discussion of the limitations of these models.

However, I have other concerns. I think that there is no clear takeaway, mainly because of the problematic study design and because of the omission of many necessary details. More detailed comments below:

* Section 1: The introduction should say more about the findings
* Section 1: A discussion of the limitations of LLMs is needed in related work. What happens if the LLM hallucinates, for example?
* Section 2.1: How many papers are available on OpenReview for ICLR 2023? Exactly 4956? How did you choose ICLR instead of any other conference?
* Section 2.1: How do the authors deal with figures in papers? Figures are important and essential to any review process.
* Section 2.3: Who are these researchers? How many of them? How were they recruited?
* Section 2.3: If you are providing annotators with six reviews in total, how can 50% be random? The chance of picking correctly the three reviews out of all six reviews are way lower than 50%. To me, ~60% accuracy means that it’s very easy to spot LLM-generated reviews and the process is not blind.
* Section 2.5: More details on how these errors are introduced are needed.
* Section 3: “Initially this paper received a recommendation score of 5 by OpenReviewer. Subsequently, we addressed the weaknesses, and improved, receiving a recommendation score of 7. “ What does this mean? Did the authors submit their own paper (without these sentences) to the submission site?
* Section 3: I’m still unclear about how P5 is introduced. In Appendix B, P5 is the answer.
* Section 3.1: What’s the takeaway here? More discussion is needed about why looking at how each P* influences ratings is important. It’s unclear to the reader.
* Section 3.2: We cannot use conference acceptance decisions as ground truth. These decisions are noisy. Thus, I don’t know what I should get out of the fact that for a couple of papers LLM and human ratings did not agree. I also don’t understand what the takeaway from Table 4 is. A scatterplot of human vs. LLM ratings would be more useful.
* An interesting question is whether the quality varies across tracks. Have the authors looked into this?
* Sections 3.3 and 3.4: More discussion is needed.
* Figure 3: Meta’s OPT-7.6B is mentioned only here. It should be mentioned before.

About the style:
* use \citep in place of \citet when appropriate
* The bars on the right of Figure 7 are redundant.
* There are some repetitions, e.g., 2040 being repeated twice in Section 2.1.
* Plots need to be improved. Font size should be large enough.

**Questions:**

Mentioned above.

---

### Official Review · Reviewer_8sHm · 2023-10-30

**Soundness:** 2 fair
**Presentation:** 3 good
**Contribution:** 2 fair
**Rating:** 5
**Confidence:** 3

**Summary:**

This paper explores the use of large language models (LLMs) like GPT for reviewing research papers. It presents a proof-of-concept that demonstrates LLMs can provide fast and consistently high-quality reviews. However, challenges such as misuse, inflated scores, and limited prompt length exist. To address these issues, the study proposes solutions like LLM watermarking, long-context windows, and blind human evaluations. The goal is to integrate these findings into an open online service, OpenReviewer, to continually refine and improve the review process with scalable human feedback.

**Strengths:**

The paper is easy to follow, it studies an interesting problem (LLM-based paper review), provides a solid proof-of concept study with insights.

**Weaknesses:**

Albeit as a solid empirical study paper, I do not see much technical novelty and am not sure if ICLR is the appropriate venue for this paper.

**Questions:**

It might help to consider another use cases of openreviewer as a tool, i.e., as part of the review decision process, including desk reject (might be safer) and as a denoiser (e.g., a prior that can be combined with human scores).

You should also report the randomness of LLM-based score for the same paper.

---

### Official Review · Reviewer_Ndo8 · 2023-11-02

**Soundness:** 2 fair
**Presentation:** 1 poor
**Contribution:** 2 fair
**Rating:** 3
**Confidence:** 3

**Summary:**

The author(s) systematically investigates the feasibility and some existing drawbacks of using LLM to write reviews of research papers. To this end, the paper proposes to mitigate the problem by marking LLM-generated reviews with LLM watermarks. Furthermore, this paper provides an online service called OpenReviewer, which serves as a convenient and effective tool for the paper reviewing process.

**Strengths:**

The paper introduces an online system in use that promotes community development. At the same time, it is the first paper to propose and introduce LLM for writing reviews for research papers.

**Weaknesses:**

1. The paper is poorly written, and there are a lot of unclear expressions. For instance, in the section on error and shortcoming detection, the method does not clearly explain how the comparison is carried out. In the experimental part of error and shortcoming detection, only a table is listed without any analysis, and it needs to be clarified what the specific meaning of the score in the experimental results is. Overall, many parts of the paper have this kind of problem.
2. The analysis of existing shortcomings is insufficiently addressed. The shortcomings mentioned in the Abstract include the risk of misuse, inflated review scores, overconfident ratings, skewed score distributions, and limited prompt length. For instance, the risk of misuse is addressed by adding a watermark, but the paper states that this watermark can also be removed, so the misuse issue is not resolved in practice. As for the limited prompt length, the paper does not provide a solution. All we see is long-context windows, implying that all available text is added to the prompt.
3. The experimental evaluation is not really convincing, as it was conducted only on one dataset from the ICLR conference. In addition, the proposed method mainly involves prompt design, and the contribution is not significant enough and needs to be condensed again.

**Questions:**

1. The amount of data mentioned in the paper is confusing. My understanding is that only papers existing on arXiv are used. In section 2.2, 10% of the dataset is used to let LLM write reviews. Then does section 2.3 also mean a further 10% on the basis of the 10% data in section 2.2? But section 3.2 mentions 5%, which is chaotic. Please provide more specific amounts of data used in each part of the evaluation.
2. Please elaborate on how to modify the paper in the section on "error and shortcoming detection"? Also, how did this work compare the paper's reviews before and after modification, and how to score it? Finally, what conclusions can be drawn from the experimental results in the "error and shortcoming detection" section?
3. Are there any advantages or innovative points about the watermark in Section 2.4?
4. Section 2.6 mainly lists points that need improvement. Has there been any experimental verification of the content?

**Details Of Ethics Concerns:**

No ethical issues